EMBO
Molecular Medicine

# IFNα gene/cell therapy curbs colorectal cancer colonization of the liver by acting on the hepatic microenvironment

Mario Catarinella[1,2], Andrea Monestiroli[1], Giulia Escobar[2,3], Amleto Fiocchi[1], Ngoc Lan Tran[1], Roberto Aiolfi[1,†], Paolo Marra[2,4,5], Antonio Esposito[2,4,5], Federica Cipriani[6], Luca Aldrighetti[6], Matteo Iannacone[1,2,5], Luigi Naldini[2,3], Luca G Guidotti[1] & Giovanni Sitia[1,*]

## Abstract

Colorectal cancer (CRC) metastatic dissemination to the liver is one of the most life-threatening malignancies in humans and represents the leading cause of CRC-related mortality. Herein, we adopted a gene transfer strategy into mouse hematopoietic stem/progenitor cells to generate immune-competent mice in which TEMs—a subset of Tie2[+] monocytes/macrophages found at peritumoral sites—express interferon-alpha (IFNα), a pleiotropic cytokine with anti-tumor effects. Utilizing this strategy in mouse models of CRC liver metastasis, we show that TEMs accumulate in the proximity of hepatic metastatic areas and that TEM-mediated delivery of IFNα inhibits tumor growth when administered prior to metastasis challenge as well as on established hepatic lesions, improving overall survival. Further analyses unveiled that local delivery of IFNα does not inhibit homing but limits the early phases of hepatic CRC cell expansion by acting on the radio-resistant hepatic microenvironment. TEM-mediated IFNα expression was not associated with systemic side effects, hematopoietic toxicity, or inability to respond to a virus challenge. Along with the notion that TEMs were detected in the proximity of CRC metastases in human livers, these results raise the possibility to employ similar gene/cell therapies as tumor site-specific drug-delivery strategies in patients with CRC.

**Keywords** colorectal cancer; gene therapy; interferon-alpha; liver metastases; tumor microenvironment

**Subject Categories** Cancer; Genetics, Gene Therapy & Genetic Disease; Immunology

## Introduction

Colorectal cancer (CRC) is one of the most common malignancies in humans and one of the leading causes of cancer-related deaths worldwide (Ferlay *et al*, 2013; American Cancer Society, 2014). Most of these deaths relate to the presence and progression of liver CRC metastases (Cunningham *et al*, 2010) and, therefore, there is a pressing need to develop more effective therapies.

Interferon-alpha (IFNα) is a pleiotropic cytokine that can impair cancer growth by directly acting as cytostatic factor on transformed cells and by negatively or positively regulating pro-tumorigenic and anti-tumorigenic processes such as angiogenesis or immunity, respectively (Pfeffer *et al*, 1998; Gough *et al*, 2012). For these reasons, IFNα has been used clinically as anti-tumor agent in different types of cancer, including CRC (Link *et al*, 2005; Wang *et al*, 2011). However, the relative inefficiency of anti-tumor therapies based on systemic IFNα administration is thought to mainly reflect the relative inability of such therapies to target effective IFNα doses to cancer sites without reaching dose-limiting toxicity (Link *et al*, 2005). To address this issue, we have employed a gene/cell-therapy approach of IFNα targeted delivery. The strategy is based on the notion that a small subset of monocytes/macrophages expressing the angiopoietin receptor Tie2 (defined as Tie2[+] monocytes/macrophages or TEMs) is recruited peritumorally in response also to hypoxic stimuli to support tumor vessel formation (De Palma *et al*, 2005; Mazzieri *et al*, 2011; Matsubara *et al*, 2013).

Taking advantage of this, we engineered mouse hematopoietic stem/progenitor cells (HSPCs) to give rise to TEMs selectively expressing an IFNα transgene allowing the release of this cytokine at therapeutic doses directly at liver tumor sites. Utilizing this approach, we showed in mouse models of CRC liver metastases that the local delivery of IFNα by TEMs exerts a potent anti-tumor

1 Division of Immunology, Transplantation and Infectious Diseases, IRCCS San Raffaele Scientific Institute, Milan, Italy
2 Vita-Salute San Raffaele University, Milan, Italy
3 Angiogenesis and Tumor Targeting Research Unit and San Raffaele Telethon Institute for Gene Therapy, IRCCS San Raffaele Scientific Institute, Milan, Italy
4 Department of Radiology, IRCCS San Raffaele Scientific Institute, Milan, Italy
5 Experimental Imaging Center, IRCCS San Raffaele Scientific Institute, Milan, Italy
6 Hepatopancreatobiliary Surgery Unit, IRCCS San Raffaele Hospital, Milan, Italy
  *Corresponding author. Tel: +39 02 2643 4956; Fax: +39 02 2643 6822; E-mail: sitia.giovanni@hsr.it
  †Present address: Department of Molecular and Experimental Medicine, The Scripps Research Institute, La Jolla, CA, USA

activity without inducing systemic side effects or hematopoietic toxicity and without altering the host immune capacity to respond to a virus challenge.

Together with the notion that TEMs were found to accumulate preferentially in the proximity of CRC metastases in the human liver, these results indicate that administering autologous genetically engineered HSPCs leading to intrahepatic delivery of IFNα by TEMs may represent a novel strategy to treat patients with CRC.

# Results

## TEMs originating from transplanted HSPCs home to the liver and gather in close proximity to hepatic CRC metastases

HSPCs isolated from the bone marrow (BM) of CB6 donor mice (H-2[bxd] F1 hybrids of C57BL/6 x BALB/c) were transduced in vitro with lentiviral vectors (LVs) granting the expression of either GFP or the murine Ifna1 gene to a subset of differentiating monocytes/macrophages expressing the angiopoietin-2 receptor Tie2 (Mazzieri et al, 2011). This was achieved by combining Tie2/Tek transcription regulatory elements and microRNA-mediated control as previously described (Escobar et al, 2014). Transduced HSPCs were transplanted into irradiated CB6 littermates defined as Tie2-GFP or Tie2-IFNα mice, respectively (see a schematic representation of the experimental strategy in Fig 1A). A group of animals transplanted with non-transduced HSPCs (defined as Mock) was utilized as additional control. Blood tests performed 7–10 weeks after HPSC transplantation revealed no significant differences in basic hematological values or in specific leukocyte subset counts among Mock or Tie2-GFP control mice and Tie2-IFNα mice (Appendix Fig S1A and B). Average vector copy numbers (VCN) per genome were measured to monitor HSPC transduction efficiency. Consistent with previous findings (De Palma et al, 2008; Escobar et al, 2014), Tie2-GFP mice (VCN = $3.21 \pm 0.16$) displayed a percentage of circulating GFP$^+$ TEMs (identified as 7AAD$^-$/CD45$^+$/CD11b$^+$/Ly6C$^+$/Ly6G$^-$/GFP$^+$ cells) that was about 0.5% of total white blood cells (WBCs; Appendix Fig S1C), suggesting by inference that a similar percentage of circulating IFNα$^+$ TEMs was present in Tie2-IFNα mice.

About 2 months after HSPC transplantation, Mock/Tie2-GFP control mice and Tie2-IFNα mice were injected intrasplenically with either the CRC cell line CT26 [H-2$^d$, BALB/c derived (Brattain et al, 1980)] or with the CRC cell line MC38 [H-2$^b$, C57BL/6 derived (Rosenberg et al, 1986)]. Note that CT26 and MC38 cell lines were found to be sensitive to the anti-proliferative effect of recombinant IFNα in vitro (Fig EV1A). To avoid intrasplenic tumor growth, the spleen was removed few minutes post-injection, thus allowing to define the impact of Tie2-IFNα approach on the growth of CRC cells that have reached the liver. Of note, the intrinsic intrahepatic behavior of CT26 and MC38 cells differed; indeed, the injection of a 10-fold different cell dose ($5 \times 10^3$ CT26 cells/mouse or $5 \times 10^4$ MC38 cells/mouse) into matched recipients resulted in almost identical survival curves (Fig EV1B).

Intrahepatic TEM identification and localization were assessed in Tie2-GFP mice at different times following injections with either saline (NaCl) or CT26 cells ($5 \times 10^3$ cells/mouse). Flow cytometric analyses of leukocytes isolated from the liver of NaCl-injected mice

indicated that a small number of 7AAD$^-$/CD45$^+$/CD11b$^+$/CD11c$^-$/GFP$^+$ TEMs (about 2% of the total intrahepatic leukocytes [IHLs], Appendix Fig S1D) is present in the organ independently of CRC cell injection (Fig 1B). By days 25 and 35 post-CRC cell injection—time points at which liver lesions are macroscopically and microscopically evident (Fig 1C)—the number of hepatic TEMs detected by flow cytometry increased (Fig 1B), and this occurred concomitantly with a commensurate increase in hepatic Tie2-driven GFP mRNA expression (Fig 1D). These findings are consistent with confocal microscopy results where the staining with antibodies specific for macrophage mannose receptor (MMR), F4/80, and GFP [3 markers previously utilized to identify TEMs-GFP in tissue (Pucci et al, 2009)] showed that few TEMs are scattered throughout the liver lobule of NaCl-injected mice (Fig 1E, top panels) and that TEMs gather in close proximity to CRC metastases (Fig 1E, bottom panels). Taken together, the results indicate that TEMs reach the liver after HSPC transplantation and that the intrahepatic number of TEMs as well as the expression of the transgene they deliver increases nearby hepatic metastatic lesions expanding in the organ.

## IFNα delivery by intrahepatic TEMs safely inhibits the growth of CRC cells into the liver

Next, Mock mice, Tie2-GFP mice (VCN = $1.46 \pm 0.15$), and Tie2-IFNα mice (VCN = $0.49 \pm 0.02$) were injected intrasplenically with $5 \times 10^3$ CT26 cells and followed individually by magnetic resonance imaging (MRI) at multiple times post-injection. By day 14, 55% of Mock/Tie2-GFP mice showed MRI-detectable lesions in the liver, while none of the Tie2-IFNα mice did (Fig 2A and B and Movies EV1, EV2, and EV3). The percentage of Mock/Tie2-GFP mice bearing disease increased to 90% and 100% by days 21 and 33, respectively, with liver lesions that—increasing in volume over time—triggered death or imposed humane euthanization of the animals (Fig 2A [left panels], B and C, and Movie EV1). Notably, 67% of Tie2-IFNα mice did not show evidence of hepatic metastases for the entire duration of the experiment (Fig 2A [center panels], B and C, and Movie EV2), and the small lesions that were detected by day 21 post-injection in 33% of the remaining Tie2-IFNα mice eventually regressed, reaching complete remission by day 54 (Fig 2A [right panels], B and C, and Movie EV3). Consistent with previous findings (De Palma et al, 2008; Escobar et al, 2014), Tie2-IFNα mice euthanized at day 25 post-injection showed a significant induction of Oas1 and to a lesser extent Irf7—two IFNα-inducible genes (Honda et al, 2005; De Palma et al, 2008)—in the metastatic liver but not in other organs such as brain or kidney (Fig 2D and E). These findings indicate that Tie2-IFNα mice are protected from the hepatic colonization of CRC cells, which upon arrival and engraftment in the liver sinusoids either do not further grow or regress after the initial establishment of few metastatic foci.

Hematopoietic stem/progenitor cells transplantation leading to stable albeit lineage-selective IFNα expression may alter the host's capacity to mount effective immune responses, as the induction of this pleiotropic cytokine is known to impact generation and function of various immune cells (Stark et al, 1998). To this end, the same Tie2-IFNα mice above mentioned (described in Fig 2A–C) were infected systemically 84 days post-CRC injection with a low

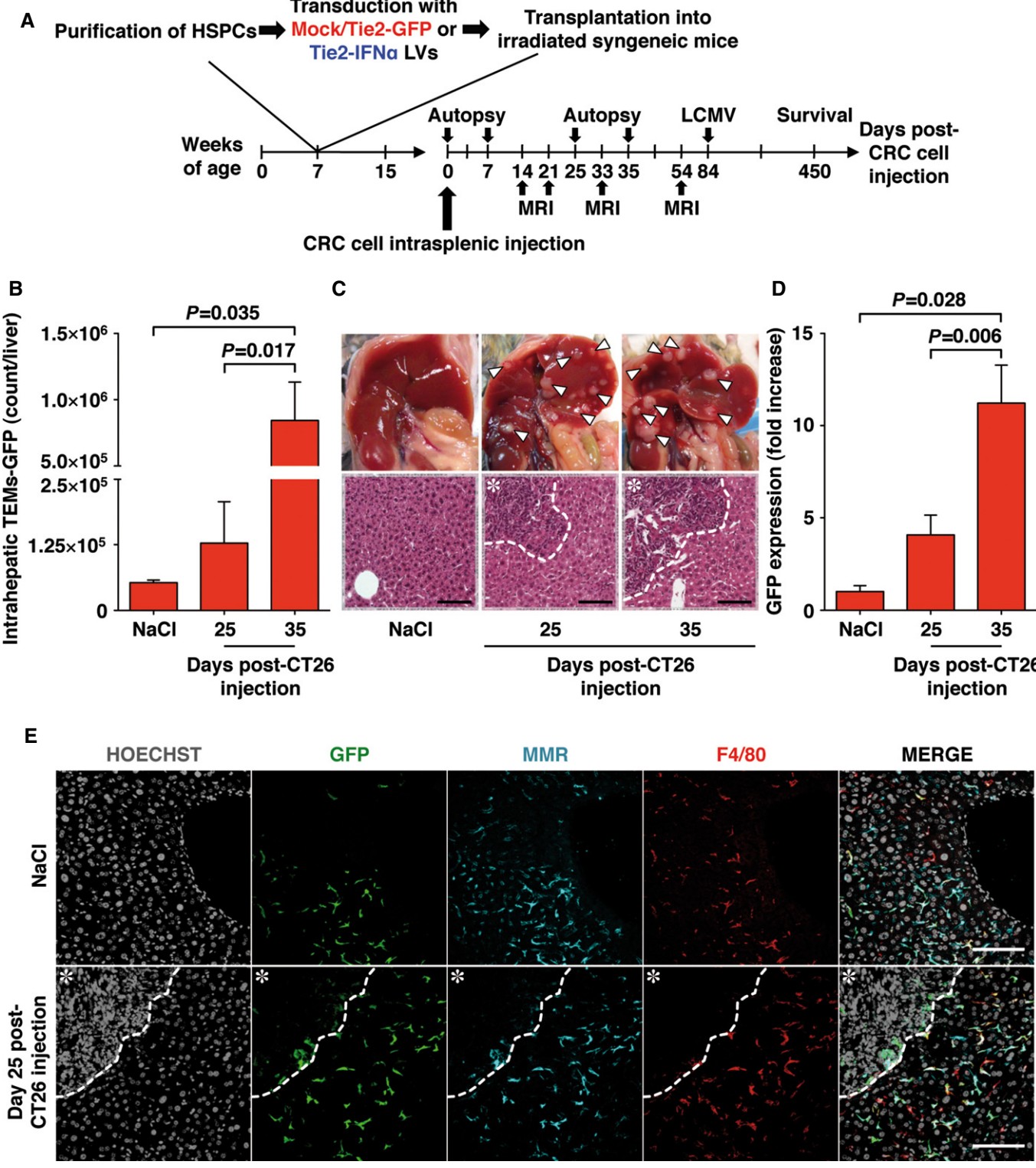

**Figure 1.**

dose (200 pfu/mouse) of lymphocytic choriomeningitis virus (LCMV), a non-cytopathic mouse pathogen known to induce a transient IFN-dependent BM aplasia followed by a robust CD8[+] T-cell response aimed at viral clearance (Binder *et al*, 1997; Iannacone *et al*, 2008). Given that no Mock or Tie2-GFP mice were still

alive at this time, Sham mice (i.e., mice transplanted with non-transduced HSPCs and intrasplenically injected with saline thereafter) or age-matched non-transplanted CB6 mice (CTRL) were used as controls. Transitory reduction in circulating WBCs and PLTs [attesting the BM aplastic response (Iannacone *et al*, 2008)]

Figure 1.  Characterization of TEMs in CRC liver metastasis mouse model.

A    Schematic representation of the experimental procedure.
B    IHLs were isolated from the liver of Tie2-GFP mice intrasplenically injected with NaCl ($n$ = 3) or with $5 \times 10^3$ CT26 at the specified time points (day 25 post-injection $n$ = 6, day 35 post-injection $n$ = 5). The number of GFP⁺ TEMs was estimated by flow cytometry as 7AAD⁻/CD45⁺/CD11b⁺/CD11c⁻/GFP⁺ cells per liver; data pooled from three independent experiments; mean values are shown; error bars indicate SEM; $P$-values were calculated by Mann–Whitney $U$-test.
C    Representative images (top panels) and corresponding H&E micrographs (bottom panels) of the liver from control (NaCl injected, left panels) or Tie2-GFP mice 25 and 35 days (center and right panels, respectively) post-intrasplenic injection of $5 \times 10^3$ CT26. Large metastatic foci in the liver of mice that received CT26 cells are indicated (arrowheads). The dashed line in the H&E panels identifies the metastasis margin; ✱ = CRC metastatic area; scale bars, 100 μm.
D    Liver RNA was extracted from mice transplanted and injected as described in (B) (NaCl $n$ = 4; day 25 $n$ = 7; day 35 $n$ = 4). The average GFP expression value of NaCl-injected mice estimated by quantitative real-time PCR was set to 1 and utilized as reference to calculate the fold increase values of CT26-injected mice; data pooled from three independent experiments; mean values are shown; error bars indicate SEM; $P$-values were calculated by Mann–Whitney $U$-test.
E    Confocal immunofluorescence images of representative liver sections from Tie2-GFP mice that were injected intrasplenically with either NaCl (upper panels) or $5 \times 10^3$ CT26 (bottom panels, 25 days post-injection). Note that TEMs (identified as GFP⁺, MMR⁺ and F4/80⁺ cells) gather in the proximity of CRC metastatic foci (identified by the dashed lines); ✱ = CRC metastatic area; scale bars, 100 μm.

(Appendix Fig S2) and presence at day 8 post-infection of relatively high numbers of circulating LCMV-specific effector CD8⁺ T cells [attesting LCMV-specific immunity (Iannacone *et al*, 2008)] were observed at comparable levels in both groups of animals (Fig 2F). This indicates that HSPC transplantation targeting TEM-mediated IFNα expression did not alter the host capacity and modality to respond to a virus challenge.

With the exception of one Tie2-IFNα mouse that died at day 438 without evidence of hepatic tumors, all of these Sham and Tie2-IFNα mice remained alive and healthy until day 450 post-CRC cell injection (Fig 2G). Notably, even at this late time point, Tie2-IFNα mice displayed basic hematological values (Fig EV2A) and hepatic morphology (Fig EV2B) comparable to those of Sham controls, with no signs of tumor relapse.

Lastly, to independently validate the anti-metastatic effect of this strategy and to exclude the possibility that the observed phenotype was restricted to a specific tumor cell line, we intrasplenically injected a higher dose ($5 \times 10^4$ cells/mouse) of MC38 cells into either Tie2-GFP mice or Tie2-IFNα mice transplanted as previously described. Importantly, also this different experimental setting,

resulted in delayed tumor appearance, reduced tumor volume and prolonged survival of Tie2-IFNα mice (Fig EV3A–C), indicating that the anti-tumor effect of our approach extends to this other CRC tumor cell line as well.

## IFNα delivery by intrahepatic TEMs impairs the early stages of intrahepatic tumor development

To assess the kinetics of hepatic IFNα release during the first week of CRC cell injection and to investigate how this process may affect the early stages of tumor development, cohorts of Tie2-GFP mice (VCN = 2.94 ± 0.19) and Tie2-IFNα mice (VCN = 0.94 ± 0.10) were euthanized at different times (5 min, day 3 or day 7) after intrasplenic injection of either NaCl or CT26 cells. Unambiguous identification and quantification of CRC cells at these early time points required the injection of higher cell numbers ($5 \times 10^5$ CT26 cells/mouse) and the adaptation of molecular and immunohisto-chemical analyses detecting a CT26 cell-specific marker (achieved by stably expressing the RFP reporter gene into CT26 cells). Analyses pertaining to IFNα release in the liver were performed by

Figure 2.  Tie2-IFNα mice display reduced tumor burden and prolonged survival.

A    Contrast-enhanced magnetic resonance imaging (MRI) of the liver from representative Tie2-GFP mice (red frame) or Tie2-IFNα (blue frame) that were intrasplenically injected with $5 \times 10^3$ CT26. Red arrows identify CRC liver metastases of representative z-sections. Tumors were characterized as hypointense and slightly hyperintense regions in T1- and T2-weighted sequences, respectively. Each panel refers to a single mouse analyzed at different time points; n.a., not assessed, refers to a mouse euthanized before the specified time point; scale bars, 5 mm.
B    Percentage of mice bearing at least one CRC liver metastasis estimated by MRI analysis. Mice were treated as described in (A). Mock/Tie2-GFP $n$ = 11, Tie2-IFNα $n$ = 9; the oblique black line pattern within the red columns depicts the percentage of mice euthanized or that died before the indicated time point; $P$-values were calculated by Fisher's exact test.
C    Tumor volume quantification measured by MRI analysis of the mice described in (B). Each symbol corresponds to the same mouse analyzed at different time points; horizontal bars, mean values; note that Mock/Tie2-GFP mice were euthanized or died before day 54; $P$-values were calculated by Mann–Whitney $U$-test.
D, E  Quantitative real-time PCR analyses of the relative expression levels of the interferon-inducible genes *Oas1* (D) and *Irf7* (E) within brain ($n$ = 4, $n$ = 2), kidney ($n$ = 4, $n$ = 3), and metastatic liver ($n$ = 4, $n$ = 7) of Tie2-GFP or Tie2-IFNα mice euthanized 25 days post-intrasplenic injection of $5 \times 10^3$ CT26. The basal expression of *Oas1* and *Irf7* estimated in brain of control mice (i.e., Tie2-GFP injected with saline) was set to 1 and utilized to calculate the fold increase values depicted. Mean values are shown; error bars indicate SEM; $P$-values were calculated by Mann–Whitney $U$-test.
F    Eighty-four or 54 days post-NaCl or $5 \times 10^3$ CT26 intrasplenic injection, Sham/CTRL ($n$ = 5) or Tie2-IFNα mice ($n$ = 7) were infected with LCMV (Armstrong strain, 200 pfu intraperitoneally injected). Eight days post-infection, white blood cells were isolated and analyzed for LCMV-specific CD8⁺ T-cell response. Left panel: percentage of total CD8⁺ T cells; middle panel: percentage of CD8⁺/NP118⁺ T cells (NP118, recombinant dimeric H-2d/Ig fusion protein complexed with the immune-dominant H-2ᵈ-restricted LCMV NP118-126 peptide); right panel: percentage of CD8⁺/IFNγ⁺ T cells after *in vitro* stimulation with virus-specific H-2ᵈ-restricted peptide (NP118-126); data pooled from two independent experiments; mean values are shown; error bars indicate SEM; differences were not statistically significant by unpaired Student's $t$-test.
G    Kaplan–Meier survival curves of the indicated groups of mice described in (A). Sham ($n$ = 3), Mock(wt)/Tie2-GFP ($n$ = 11), Tie2-IFNα ($n$ = 9); data pooled from three independent experiments; $P$ = 0.001 by log-rank/Mantel–Cox test. The inset images show representative macroscopic photographs of the metastatic progression in the liver of a control animal (red frame: Mock/Tie2-GFP, day 35 post-CT26 injection) opposed to the lack of lesions in Tie2-IFNα-treated mice at later time points (blue frame: day 450 post-CT26 injection).

Figure 2.

monitoring the hepatic expression of *Irf7*, a prototypical IFNα-inducible gene (Honda *et al*, 2005).

Consistent with the notion that small numbers of TEMs are detectable in the liver of HSPC-transplanted mice independently of CRC cell injection (Fig 1B, D, and E, top panels), the hepatic content of *Irf7* mRNA in Tie2-GFP mice or Tie2-IFNα mice that were injected with NaCl diverged, with the latter animals showing a larger than threefold increase over basal (Fig 3A). Irrespective of this, both immunohistochemical and molecular analyses detecting hepatic RFP expression showed that similar numbers of CT26-RFP cells reached the liver of Tie2-GFP and Tie2-IFNα mice by 5 min post-injection (Fig 3B [top panels] and C, and Appendix Fig S3A), indicating that the higher IFNα levels detected in Tie2-IFNα mice did not affect the capacity of CRC cells to initially engraft the liver parenchyma.

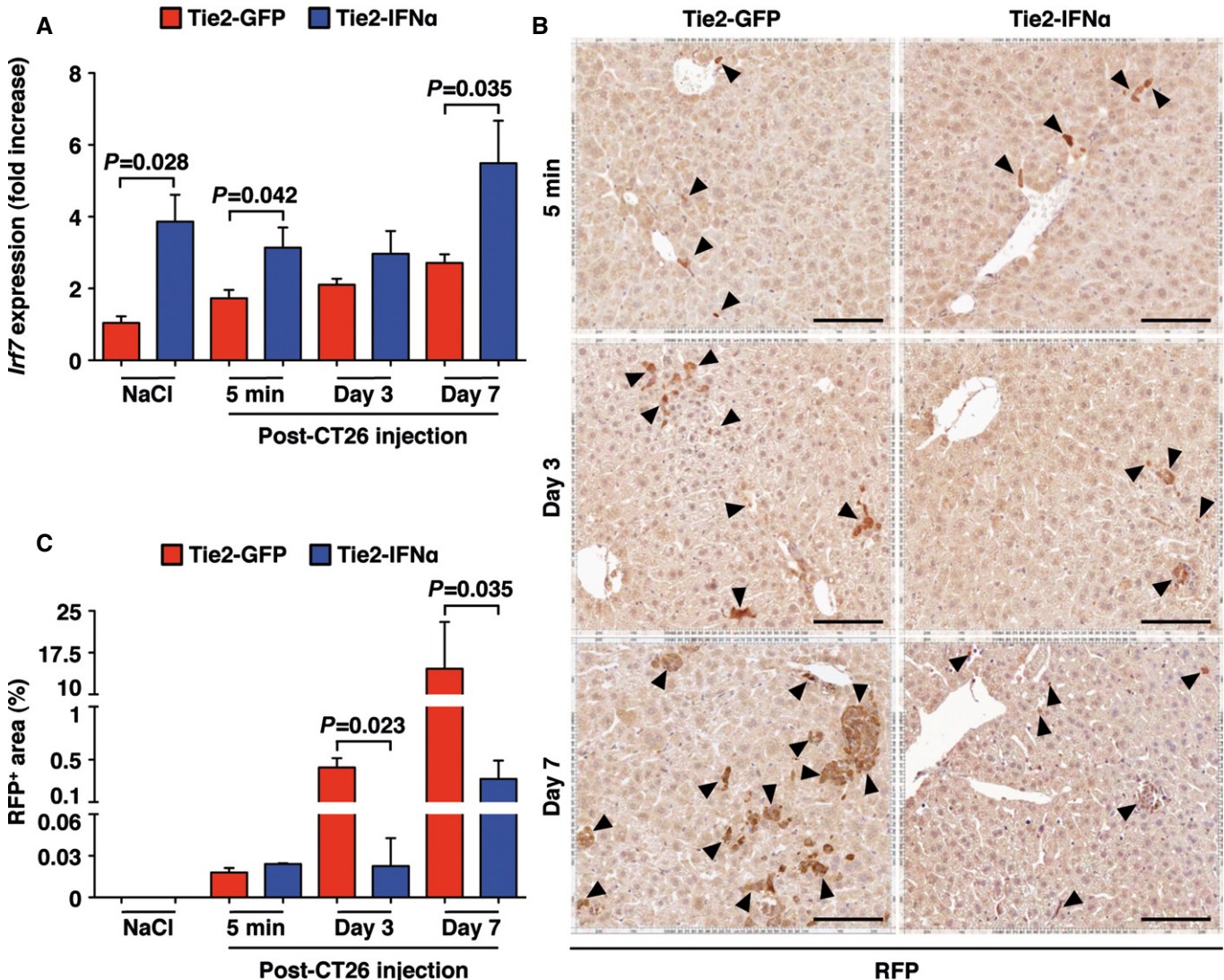

**Figure 3. CT26-RFP⁺ CRC cell colonization of the liver.**

A   Quantitative real-time PCR analyses of the relative expression levels of the interferon-inducible gene *Irf7* within the liver of Tie2-GFP or Tie2-IFNα mice that were intrasplenically injected with either NaCl ($n = 4$ and $n = 4$, respectively) or $5 \times 10^5$ CT26-RFP and euthanized 5 min ($n = 9$, $n = 5$), 3 days ($n = 6$, $n = 3$) and 7 days ($n = 5$, $n = 3$) thereafter. The basal expression of *Irf7* estimated in control mice (i.e., Tie2-GFP injected with saline) was set to 1 and utilized to calculate the fold increase values observed at the following time points post-injection. Data pooled from four independent experiments; mean values are shown; error bars indicate SEM; *P*-values were calculated by Mann–Whitney *U*-test. The increase in *Irf7* expression levels from the liver of Tie2-GFP mice was statistically significant ($P = 0.004$ by one-way ANOVA test, not reported on graph).

B   Representative RFP immunostaining from the liver of Tie2-GFP mice (left panels) or Tie2-IFNα mice (right panels) at the indicated time points after $5 \times 10^5$ CT26-RFP intrasplenic injection. Arrowheads highlight single or clustered RFP-positive cells; scale bars, 100 μm.

C   Immunostaining quantification of hepatic CT26-RFP arrival (5 min post-injection: Tie2-GFP $n = 6$, Tie2-IFNα $n = 2$) and expansion (day 3 post-injection: Tie2-GFP $n = 6$, Tie2-IFNα $n = 3$; day 7 post-injection: Tie2-GFP $n = 5$, Tie2-IFNα $n = 3$) in the liver of Tie2-GFP mice or Tie2-IFNα mice that were injected with $5 \times 10^5$ CT26-RFP as described in (B). Data pooled from two independent experiments; mean values are shown; error bars indicate SEM; *P*-values were calculated by Mann–Whitney *U*-test. The increased percent of RFP⁺ areas in the liver of Tie2-GFP mice was statistically significant ($P = 0.0009$ by one-way ANOVA test, not reported on graph).

Starting already by day 3, however, the growing behavior of CRC cells differed substantially in the 2 cohorts of animals (Fig 3B, middle panels). Indeed, the hepatic CT26 cell content at this time point increased more than 10-folds in Tie2-GFP mice, while it remained basically unaltered in Tie2-IFNα mice (Fig 3C). By day 7, the different growing rate of CRC cells in the 2 cohorts of animals became even more evident, with the appearance of large clusters of CRC cells in the liver of Tie2-GFP mice that were not present in the liver of Tie2-IFNα mice (Fig 3B [bottom panels] and C). Lack of CRC cell growth in the latter animals was associated with *Irf7* mRNA levels that were significantly induced compared to similarly injected Tie2-GFP controls (Fig 3A). The finding that the hepatic content of *Irf7* mRNA in Tie2-GFP animals slightly increased over time (from 5 min to day 7 after CT26-RFP cell injection, Fig 3A) suggests that CRC cell expansion *per se* stimulates the release of low levels of endogenous IFNα; such levels of this cytokine, however, appear to be not sufficient to contain tumor growth. In fact, 7 days after CRC cell injection, the metastatic foci in Tie2-GFP mice were highly proliferative (as denoted by positivity to the Ki67 signal, Fig EV4A) and associated with newly formed vessels (as denoted by positivity to the CD34 signal, Fig EV4B) and with the significant upregulation of the pro-angiogenic marker *Angpt2* (Appendix Fig S3B). These results indicate that (early) intrahepatic proliferation of CRC cells in Tie2-GFP mice was related to signs of tumor-driven angiogenesis. The relatively small content of RFP$^+$ (Fig 3B [bottom panels] and C) and Ki67$^+$ (Fig EV4A) metastatic foci detected in the liver of Tie2-IFNα mice instead indicates that CRC cells expanded more slowly than those residing in the liver of Tie2-GFP mice. Notably, this effect was associated with lower numbers of CD34$^+$ vessels (Fig EV4B) and lower expression of *Angpt2* (Appendix Fig S3B). Further immunohistochemical analyses revealed a comparable increase—over NaCl-injected Tie2-GFP mice—in the number of F4/80$^+$ macrophages as well as CD3$^+$ T cells and B220$^+$ B cells in the liver of both Tie2-GFP mice and Tie2-IFNα mice at day 7 but not at day 3 post-CT26-RFP cell injection (Fig EV4C–E). This suggests that CRC cell expansion (more abundant in Tie2-GFP mice) and IFNα release (more abundant in Tie2-IFNα mice) both promoted the intrahepatic recruitment and/or expansion of immune cells, reaching at day 7 a similar overall effect. Note that at this time point, the ratio of recruited immune cells per tumor cell was much higher in Tie2-IFNα mice. All together, these results indicate that TEM-mediated delivery of IFNα to the liver effectively impairs the early stages of intrahepatic CRC growth.

## Liver radio-resistant cells are primary targets of the anti-tumor activity of IFNα

To investigate which cellular compartments are targeted by the anti-tumor activity of IFNα, we injected C57BL/6-derived MC38 CRC cells into BM chimeric animals in which only selected cell populations carry the IFNα/β receptor. Briefly, HSPCs purified from either C57BL/6 mice or IFNα/β receptor knockout mice (inbred C57BL/6, from now on indicated as IFNα/βR$^{-/-}$) were transduced *in vitro* with Tie2-GFP or Tie2-IFNα LVs. Transduced HSPCs were transplanted into lethally irradiated C57BL/6 or IFNα/βR$^{-/-}$ recipients, generating BM chimeras (see a schematic representation in Appendix Fig S4A) where the cellular compartments that could respond to IFNα are (i) CRC cells + all other cells of the body (C57BL/6-derived HSPCs transplanted into C57BL/6 recipient mice; VCN: Tie2-GFP = 2.25 ± 1.02, Tie2-IFNα = 0.56 ± 0.26); (ii) CRC cells + hematopoietic cells (C57BL/6-derived HSPCs transplanted into IFNα/βR$^{-/-}$ recipient mice; VCN: Tie2-GFP = 1.00 ± 0.08, Tie2-IFNα = 0.41 ± 0.06); and (iii) CRC cells + radio-resistant cells (IFNα/βR$^{-/-}$-derived HSPCs transplanted into C57BL/6 recipient mice; VCN: Tie2-GFP = 6.68 ± 1.31, Tie2-IFNα = 7.81 ± 0.97).

It is of note that IFNα/βR$^{-/-}$ HSPCs transduced with 40-fold less Tie2-GFP or Tie2-IFNα LVs displayed higher vector copy numbers than C57BL/6 HSPCs transduced with the same LVs (Appendix Fig S4B), suggesting a role for IFNα at inhibiting LV transduction *in vitro*. It is also of note that basic hematological values of chimeric animals transduced with either Tie2-GFP or Tie2-IFNα LVs were not significantly different (Appendix Fig S4C).

As mentioned above, the different groups of BM chimeras were intrasplenically injected with syngeneic MC38 CRC cells (5 × 10$^4$ cell/mouse) and the liver of each animal was analyzed 2 and 3 weeks later by MRI. As expected, the comparison between Tie2-GFP mice and Tie2-IFNα mice in which all cells are responsive to IFNα (C57BL/6 HSPCs into C57BL/6 mice) showed a significant reduction in overall tumor volume in the latter group at both time points analyzed, confirming once more the anti-tumor efficacy of our approach (Fig 4A [top panels] and B, and Movie EV4). A similar comparison between Tie2-GFP mice and Tie2-IFNα mice in which only CRC cells and cells of hematopoietic origin are responsive to IFNα (C57BL/6 HSPCs into IFNα/βR$^{-/-}$ mice) revealed tumor growth in both groups of animals (Fig 4A [middle panels] and B, and Movie EV5), indicating that IFNα/β receptor-mediated signaling on radio-resistant cells is necessary to confer anti-tumor activity during the early phase of liver metastasis development. The lack of effect in these Tie2-IFNα mice—where, like in the other groups, tumor cells have the potential to respond to IFNα—also argues against the possibility of a direct anti-tumor activity of IFNα on CRC cells as principal anti-tumor mechanism. Similar to the situation where all cells are responsive to IFNα, the comparison between Tie2-GFP mice and Tie2-IFNα mice bearing the IFNα/β receptor only on CRC cells and radio-resistant cells (IFNα/βR$^{-/-}$ HSPCs into C57BL/6 mice) indicated a reduction in the overall volume of tumor lesions in the latter group (Fig 4A [bottom panels] and B, and Movie EV6). These data indicate that radio-resistant hepatic cells are primary targets of IFNα necessary for the Tie2-IFNα-based therapy to perform its full anti-metastatic potential, which likely involves additional anti-tumor activities mediated by BM-derived immune cells (including TEMs themselves), consistently with previous publications (De Palma *et al*, 2008; Escobar *et al*, 2014).

Among liver-resident radio-resistant cells such as endothelial cells, fibroblasts, stellate cells, hepatocytes, and Kupffer cells (KCs) (Klein *et al*, 2007), the latter represent an attractive target due to their reported capacity to contain early CRC growth within the liver (Wen *et al*, 2013). To investigate this hypothesis, 2 cohorts of C57BL/6 mice that received IFNα/βR$^{-/-}$ HSPCs transduced with either Tie2-GFP (VCN = 7.23 ± 2.16) or Tie2-IFNα (VCN = 9.18 ± 0.80) LVs were treated 4 weeks later with KC-depleting clodronate-containing liposomes (Clo-L) as described (Sitia *et al*, 2011) (see the Clo-L effect on KCs in Fig EV5A, center panel). Another 4 weeks later—a time point at which the liver was repopulated with KCs

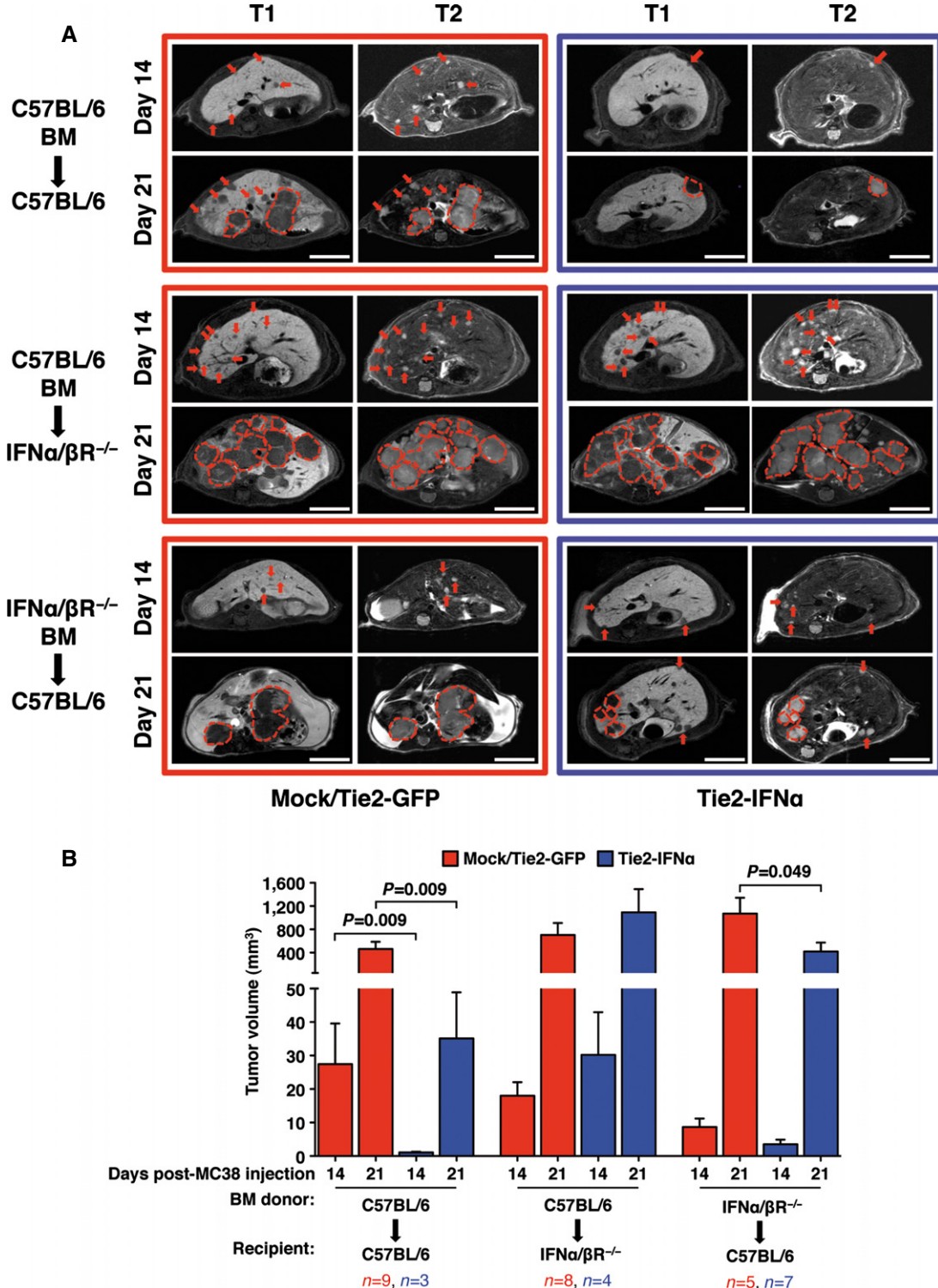

**Figure 4. Liver radio-resistant cells are primary targets of the anti-tumor activity of IFNα.**

A   Representative contrast-enhanced MRI panels depicting the liver of Mock/Tie2-GFP (red frame) or Tie2-IFNα (blue frame) chimeric mice. Each frame shows the metastatic progression 14 and 21 days post-intrasplenic injection of $5 \times 10^4$ MC38 within the indicated chimeric group. BM donors and recipient mouse strains are indicated on the left of each frame; red arrows or dashed red lines identify CRC liver metastases of representative z-sections. Tumors were identified as hypointense and slightly hyperintense regions in T1- and T2-weighted sequences, respectively; scale bars, 5 mm.

B   Tumor volume quantification (based on MRI analyses) of lesions detected in the same livers described in (A). Chimeric groups and number of mice analyzed are listed; data pooled from three independent experiments; mean values are shown; error bars indicate SEM; *P*-values were calculated by Mann–Whitney *U*-test.

(Fig EV5A, right panel)—the 2 cohorts of animals were injected intrasplenically with MC38 CRC cells ($5 \times 10^4$ cell/mouse). At this time point, previous studies show that under similar conditions KCs are of hematogenous origin (Klein *et al*, 2007) and, thus, lack the IFNα/β receptor. MRI analyses performed 14 days post-CRC cell injection revealed that the lack of IFNα/β receptor on KCs did not inhibit the capacity of IFNα to limit tumor growth (Fig EV5B and C, and Movie EV7).

All in all, the results indicate that the liver radio-resistant microenvironment is a primary target of the IFNα-dependent anti-tumor activity that contains tumor growth.

## Effective intrahepatic CRC cells growth inhibition and improved survival following Tie2-IFNα HSPC transplantation in mice with established CRC metastatic liver tumors

In order to examine the impact of Tie2-IFNα on established liver metastases, CB6 mice were injected intrahepatically (beneath the Glisson's capsule) with $5 \times 10^3$ CT26 CRC cells and transplanted 8 days later with Tie2-GFP- or Tie2-IFNα-transduced HSPCs (see a schematic representation of the experimental strategy in Fig 5A). Note that the sub-capsular injection approach was utilized to minimize CRC cell spreading within the liver, allowing mice to reconstitute their BM in the presence of established and fast-growing hepatic lesions. Four days after transplantation—a time point in which Tie2-GFP or Tie2-IFNα cells are not yet emerged from the BM (Lechman *et al*, 2012; Zonari *et al*, 2013)—the animals were subjected to the first MRI analysis, which detected no significant differences in the number of lesions per liver of the two groups of mice (average metastasis number at day 4: Tie2-GFP = $1.5 \pm 0.4$; Tie2-IFNα = $2 \pm 0.28$) and in their total tumor volume (Fig 5B and C, and Movie EV8). Follow-up MRI analyses at days 21 and 28 post-transplant revealed similar numbers of intrahepatic lesions (average metastasis number at days 21-28: Tie2-GFP = $3 \pm 0.53$; Tie2-IFNα = $3.12 \pm 0.63$); however, Tie2-IFNα mice displayed reduced volumes of CRC liver metastases when compared to those detected in Tie2-GFP mice (Fig 5B and C, and Movie EV8). This difference fell short of being statistically significant, probably because of the partial hematopoietic reconstitution at these time points (note that at day 28 after BM transplantation, peripheral WBCs were still about 30% below normal counts, Appendix Fig S5A). Of note, Tie2-IFNα mice showed a 70% reduction in the appearance of peritoneal carcinomatosis at both days 21 and 28 post-transplant (a complication of Glisson's capsule infiltration, with CRC cells spreading into the peritoneal cavity) when compared to Tie2-GFP mice (Fig 5C and D, and Movie EV8). This is consistent with the notion that TEMs accumulate nearby metastatic lesions also in this setting (30 days after transplant, Fig 5E, bottom panels), when BM reconstitution has not reached completion (Appendix Fig S5A).

Molecular analyses performed 30 days post-transplant showed an increased expression of the IFNα-inducible genes *Irf7* and *Oas1* in the liver of Tie2-IFNα mice (when compared to the hepatic expression of *Irf7* and *Oas1* in Tie2-GFP mice, Fig 5F and G). Remarkably, Tie2-IFNα mice showed a much-improved overall survival over Tie2-GFP mice (Fig 5H).

No significant differences in specific leukocyte subset counts were observed between Tie2-GFP mice (VCN = $12.85 \pm 0.96$) and Tie2-IFNα mice (VCN = $1.73 \pm 0.2$) at day 30 post-transplant (Appendix Fig S5B) with a percentage of circulating GFP$^+$ TEMs (identified as 7AAD$^-$/CD45$^+$/CD11b$^+$/Ly6C$^+$/Ly6G$^-$/GFP$^+$ cells) that reached 2% (Appendix Fig S5C) in Tie2-GFP mice. No significant differences in the number of splenic CD4$^+$ T cells or CD8$^+$ T cells were also observed between the groups of mice (Appendix Fig S5D and E). This was accompanied by a slight increase in markers of activation (e.g., CD25, CD69, and PD1) and markers of central memory differentiation in the CD8$^+$ T-cell compartment (Appendix Fig S5E). In keeping with what we previously reported (De Palma *et al*, 2008; Escobar *et al*, 2014), the results suggest that the status of immune cell activation between the two groups of mice is quite comparable, with a trend of higher CD8$^+$ T cell with an activated phenotype in Tie2-IFNα mice.

---

**Figure 5. Effective intrahepatic CRC cells growth inhibition and improved survival following Tie2-IFNα HSPC transplantation in mice with established metastatic liver tumors.**

A Schematic representation of the experimental procedure.

B Tumor volume quantification measured by MRI analysis of Tie2-GFP (*n* = 6) or Tie2-IFNα mice (*n* = 9) at the indicated time points post-transplant, one additional Tie2-IFNα mouse showed a tumor volume at day 21 of more than 820 mm$^3$ and was statistically rejected by the Grubbs' test and not further analyzed; mean values are shown; error bars indicate SEM; *P* = 0.051, by Mann–Whitney *U*-test.

C Images (top panels) and corresponding contrast-enhanced MRI (middle and bottom panels) of the liver from representative Tie2-GFP mice or Tie2-IFNα mice that were intrahepatically injected with $5 \times 10^3$ CT26. White dashed lines identify macroscopic lesions. Red and green dashed lines identify hepatic or extrahepatic CRC metastases, respectively, from representative MRI z-stacks. Tumors detected by MRI analysis appeared as hypointense and slightly hyperintense regions in T1- and T2-weighted sequences, respectively. Scale bars, 5 mm.

D Percentage of mice with peritoneal carcinomatosis (defined by the presence of multiple intra-peritoneal lesions) determined by MRI analysis from the mice described in (B). Note that while the number of intrahepatic lesions remained similar between time points, extrahepatic tumor spreading was significantly reduced in Tie2-IFNα mice. *P*-values were calculated by Fisher's exact test.

E Confocal immunofluorescence images of representative liver sections from Tie2-GFP mice at distal (upper panels) or proximal (bottom panels) areas to CRC liver metastasis, 30 days post-HSPC transplant. Note that TEMs were identified as GFP$^+$ cells and by the concomitant expression of different levels of the myeloid cell marker CD11b$^+$ as highlighted in the inset (merge panels, white arrows); the dashed line identifies the metastasis margin; ✱ = CRC metastatic area; scale bars, 100 μm; insets depict the corresponding areas identified by dashed squares in the merge panels, magnified 1.5-folds.

F, G Quantitative real-time PCR analyses of the relative expression levels of the interferon-inducible genes *Irf7* and *Oas1* within liver of Tie2-GFP (*n* = 4) or Tie2-IFNα (*n* = 4) mice euthanized 30 days post-transplant. The basal expression of *Irf7* estimated in liver of control mice (i.e., Tie2-GFP injected with saline) was set to 1 and utilized to calculate the fold increase values observed. Mean values are shown; error bars indicate SEM; *P*-values were calculated by Mann–Whitney *U*-test.

H Kaplan–Meier survival curves of the indicated groups of mice. Tie2-GFP (*n* = 3); Tie2-IFNα (*n* = 5); *P* = 0.007 by log-rank/Mantel–Cox test.

---

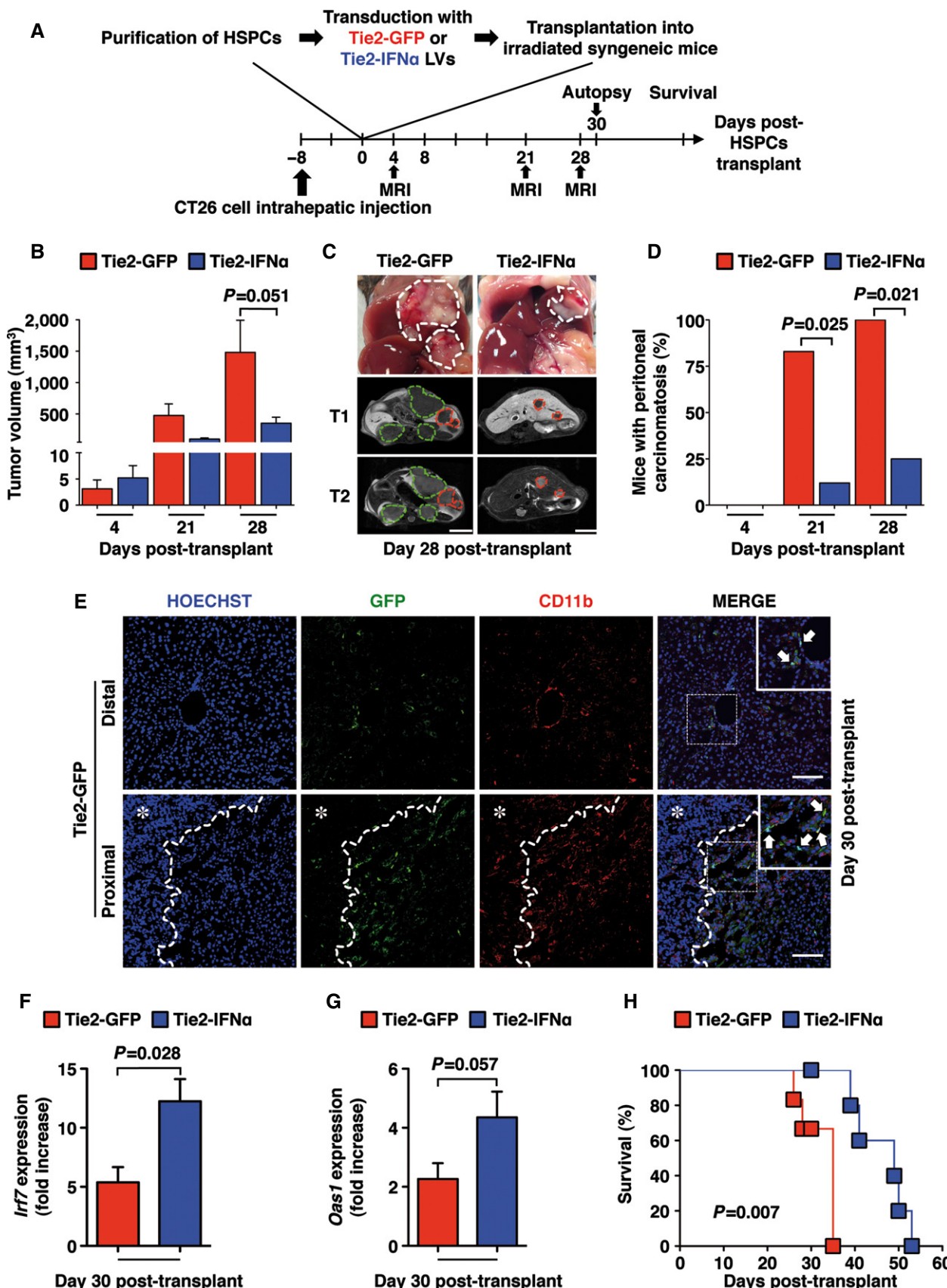

Figure 5.

**Figure 6.  TEMs preferentially accumulate in the peritumoral area of human colorectal cancer liver metastases.**

A     IHL and TEM characterization in the liver of patients described in Appendix Fig S6A. Liver samples from surgical resections (*n* = 9) were weighted and IHLs were isolated, counted and normalized over tissue weight (left panel). Flow cytometry analysis revealed that TEMs, identified as 7AAD$^-$/CD45$^+$/CD11b$^+$/CD14$^+$/Tie2$^+$ cells, are detectable within the total IHL populations and accumulate preferentially in the proximity of the metastatic lesions (right panel). Control, hepatic hemangioma; Distal, distal to CRC liver metastasis (> 1 cm from the lesion); Proximal, proximal to CRC liver metastasis (< 1 cm from the lesion); *P*-values were calculated by Wilcoxon matched pairs test.

B, C   Representative TIE2 immunohistochemistry staining from the liver of a patient with liver CRC metastases from distal (> 1 cm; left panels) or proximal (< 1 cm; right panels) sites from the CRC liver lesion. (C) Higher magnification of the panels identified by rectangles in (B). Negative controls (bottom panels) were obtained by omitting primary anti-TIE2 antibodies. TIE2 immunostaining indicates that sites that are distal from the lesion contain only TIE2$^+$ cells with an apparent endothelial morphology (characterized by an elongated appearance, arrows), while sites that are proximal to the lesion also contain additional TIE2$^+$ cells with an apparent monocyte-like morphology (characterized by a round appearance, arrowheads). The dashed line identifies the metastasis margin; ✱ = CRC metastatic area. Scale bars of upper panels, 50 μm; middle and bottom panels, 20 μm.

## TEMs preferentially accumulate in the peritumoral area of human colorectal cancer liver metastases

To characterize the presence of TEMs in the liver of patients carrying CRC metastases, liver specimens obtained from 8 patients that underwent surgical resection of CRC hepatic metastases were analyzed. One additional liver resection specimen containing a benign hepatic hemangioma (from a ninth patient) was analyzed and used as control (see Appendix Fig S6A to evaluate the various characteristics of the patients mentioned above). Immediately after liver resection, fresh tissue specimens were sampled from peritumoral areas that were either proximal to the CRC lesion edge (< 1 cm from the lesion edge) or distal from it (more than 1 cm from the lesion edge), so that the latter specimens could be considered as internal, non-tumor controls. Tissue specimens were normalized by weight and reduced to single-cell suspensions from which IHL populations were isolated, counted, and analyzed by flow cytometry. Human TEMs were identified by a 7AAD$^-$/CD45$^+$/CD11b$^+$/CD14$^+$/Tie2$^+$ profile as previously described (Murdoch *et al*, 2007; Venneri *et al*, 2007) (Appendix Fig S6B–G). It is noteworthy that almost no Tie2$^+$/CD45$^-$ cells (e.g., liver sinusoidal endothelial cells, LSECs) were detected within IHL preparations (Appendix Fig S6D). Flow cytometric analyses of these leukocyte populations infiltrating diseased human livers revealed that proximal areas contain more IHLs and more TEMs than distal areas and that only few TEMs are found in the control liver specimen (Fig 6A). Immunohistochemical analyses performed on formalin-fixed liver tissue indicated that sites that are distal from the lesion (more than 1 cm from the lesion edge) contain TIE2$^+$ cells possessing only an endothelial-like morphology (characterized by an elongated appearance, Fig 6B and C, left panels), while sites that are close to the lesion (< 100 μm from the lesion edge) contained additional TIE2$^+$ cells with a monocyte-like morphology (characterized by a round appearance, Fig 6B and C, right panels). The preferential accumulation of TEMs in areas that are proximal to CRC lesions was apparently independent of disease stage, tumor burden, or previous chemotherapeutic intervention (Appendix Fig S6A).

## Discussion

We showed herein that targeted delivery of IFNα by gene/cell therapy to the liver prevents, reduces, or reverts the growth of hepatic CRC metastases and improves overall survival in immune-competent mice. As previously described (De Palma *et al*, 2008; Escobar *et al*, 2014), targeted delivery of IFNα was achieved by engineering HSPCs to give rise to TEMs that selectively express this cytokine. Immune-competent mice carrying genetically modified TEMs and bearing hepatic CRC metastases were generated by injecting syngeneic CRC cell lines into the spleen after HSPC transplantation or by transplanting HSPCs into mice that were previously injected with syngeneic CRC cell lines directly into the liver.

We initiated our study by following longitudinally the impact that HSPC transplantation prior to CRC cell injection might have on the progression of hepatic CRC cell growth. These experiments revealed that few weeks after CRC cell injection TEMs accumulate in the proximity of hepatic CRC metastases and that TEM-mediated delivery of IFNα exerts a potent anti-tumor activity. Indeed, mice carrying IFNα-expressing TEMs showed either no hepatic lesions or metastatic foci that were much smaller than those detected in control animals, with some animals that displayed tumor regression leading to complete and permanent remission. HSPC transplantation promoting IFNα expression by TEMs caused the hepatic induction of IFNα-regulated genes, and this caused neither detectable side effects nor hematopoietic toxicity and it did not inhibit the host's capacity to respond to a virus challenge. Of note, no or little induction of IFNα-regulated genes was detected in organs that did not bear tumors.

The experiments mentioned above indicate that CRC cell growth favors the intrahepatic recruitment of TEMs which when engineered to express transgenic IFNα, display great anti-tumor potential. Additional analyses at early time points also unveiled that a small number of TEMs resides into the liver of HSPC-transplanted mice even before CRC arrival and that the local delivery of IFNα by these cells limits the initial phases of hepatic CRC cell expansion and the early formation of tumor-associated vessels.

Thus, two non-mutually exclusive hypotheses might explain how genetically engineered TEMs exert anti-tumor potential in our system. First, the basal release of IFNα by liver-resident TEMs restricts the early steps of metastatic growth. This concept is supported by the notion that IFNα-responsive genes are induced—albeit to levels only slightly higher than controls—in the normal liver of mice carrying IFNα-expressing TEMs and that patients with CRC suffering from chronic liver diseases associated with elevated levels of IFNα into the organ display reduced incidence of hepatic CRC metastases (Lieber, 1957; Iascone *et al*, 2005; Li Destri *et al*, 2013; Cai *et al*, 2014). Second, the hepatic expansion of CRC cells not initially contained by the basal release of IFNα favors the local

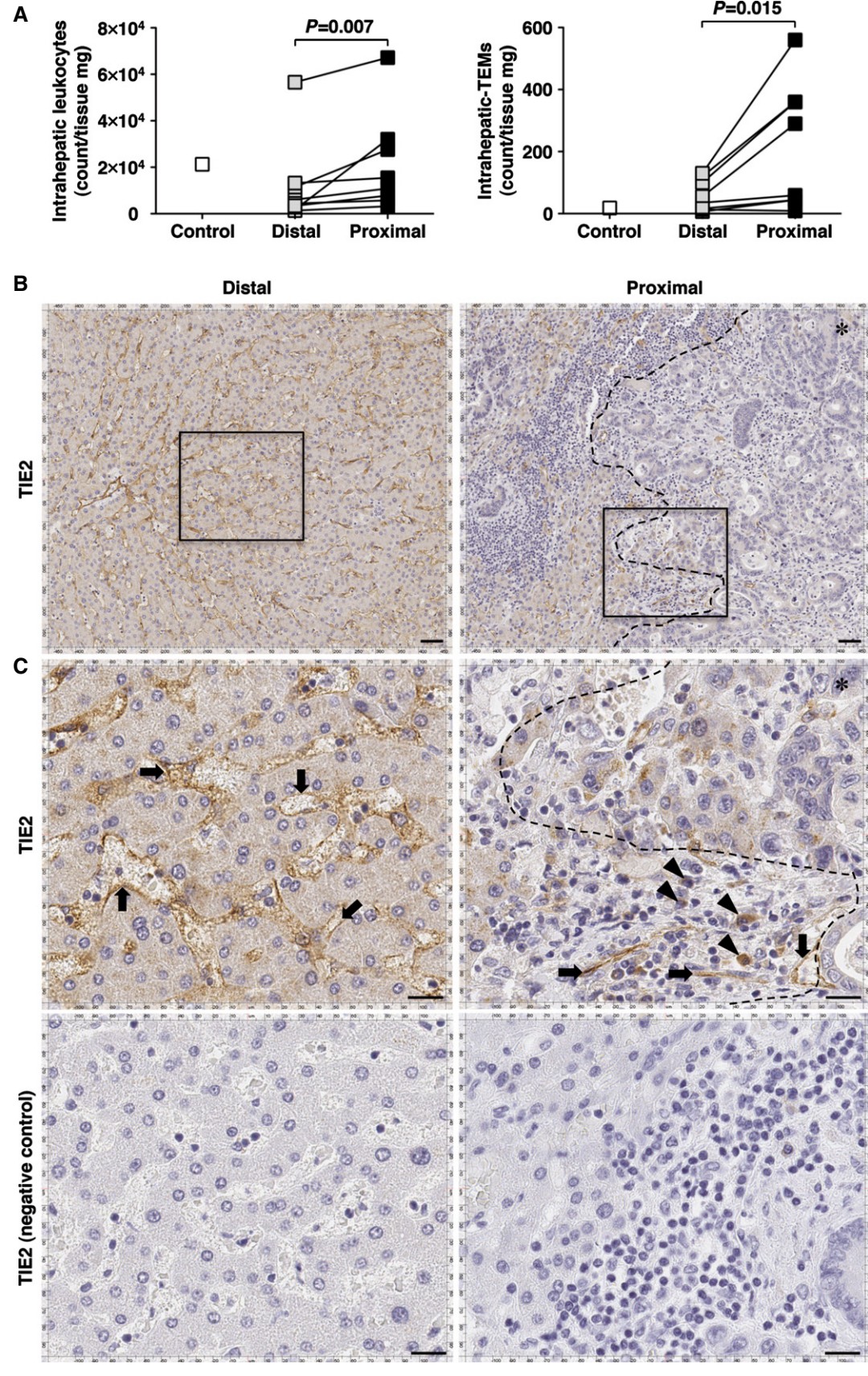

**Figure 6.**

recruitment of TEMs from blood and, therefore, triggers additional release of IFNα at tumor sites. This latter concept is supported by the notion that TEM recruitment nearby CRC lesions does occur and that the tumor growth-related increase in hepatic IFNα-responsive genes during the early phases of CRC cell expansion occurs more proportionally in mice carrying transgenic IFNα-expressing TEMs than in mice carrying GFP-expressing TEMs.

The data obtained from non-reciprocal IFNα/βR$^{-/-}$ BM chimeras then indicated that liver radio-resistant stromal cells represent a primary target of the anti-tumor activity of IFNα, in keeping with the idea that the hepatic stroma can contribute to metastatic cancer cell proliferation (Braet *et al*, 2007; Vidal-Vanaclocha, 2008; Sleeman *et al*, 2012; Spaapen *et al*, 2014). The difficulty of selectively eliminating radio-resistant liver cells other than KCs (e.g., endothelial cells, fibroblasts, stellate cells, and hepatocytes) *in vivo* coupled with the possibility that more than one cell population could be simultaneously targeted by IFNα renders impracticable to design experiments aimed at identifying which radio-resistant liver cells are ultimately targeted by this pleiotropic cytokine. Nonetheless, our findings also illustrate that CRC metastatic growth is still impaired when TEMs themselves cannot respond to IFNα (when they lack the corresponding receptor), implicating that IFNα-dependent signaling is not cell autonomous in these cells and that the anti-tumor effect observed in this setting is independent of transgenic IFNα altering TEM differentiation or activation.

Follow-up experiments in mice that were transplanted with HSPCs after being injected intrahepatically with CRC cells also demonstrated efficacy and safety of our anti-tumor approach. As before, TEMs were found to accumulate in the proximity of hepatic CRC metastases and their capacity to deliver IFNα locally was associated with improved overall survival.

Considering clinical applications, our protocol could be adopted as adjuvant therapy in patients that either harbor established metastases or are at high risk of developing metachronous liver metastases following surgical removal of primary tumors (Chuang *et al*, 2011). Moreover, the protocol could also be applied to patients that previously received radiofrequency ablation of established liver metastases, a percutaneous procedure often associated with high rates of metastatic CRC recurrence into the liver (Guenette & Dupuy, 2010). Along these lines of thinking, the enrichment of TEMs that we observed in the proximity of human CRC liver metastases suggests that these cells, once properly engineered, are best positioned to target effective IFNα doses to cancer sites. The construction of LVs inserting effective human *TIE2* and *IFNα* sequences into human Tie2 monocytes/macrophages (Escobar *et al*, 2014) and the notion that protocols of reduced-intensity autologous stem cell transplantation already entered the clinical stage (Aiuti *et al*, 2013) represent relevant steps that should facilitate the access of our strategy in the clinic, thus providing a novel approach to treat hepatic CRC metastases in humans.

# Materials and Methods

## Disease models

The mouse models of CRC liver metastases utilized in most of the experiments are CB6 mice, obtained by crossing *M. m. domesticus*

inbred C57BL/6 male mice (H-2$^b$ restricted) with *M. m. domesticus* inbred BALB/c female mice (H-2$^d$ restricted) both obtained by Charles River Laboratories, to produce H-2$^{bxd}$ F1 hybrids. IFNα/βR$^{-/-}$ mice in C57BL/6 background (obtained through the Swiss Immunological Mutant Mouse Repository, Zurich, Switzerland) were employed in selected experiments as reported in the text. Lineage-negative HSPC-enriched BM cells were isolated as previously described (Escobar *et al*, 2014) from CB6 mice, C57BL/6 mice, or IFNα/βR$^{-/-}$ mice (Appendix Supplementary Materials and Methods) according to the experimental needs. HSPCs were then transduced for 12 h with $10^8$ TU/ml of indicated lentiviral particles (IFNα/βR$^{-/-}$ HSPC-enriched cells were transduced with $2.5 \times 10^6$ TU/ml lentiviral particles instead) as described in Appendix Supplementary Materials and Methods, and intravenously injected in the tail vein of lethally irradiated (700 Rad) randomized recipient male mice ($1 \times 10^6$ cells/mouse). Eight to 10 weeks after HSPC transduction/transplantation, recipient mice were intrasplenically injected with different doses of CRC cells (either CT26, CT26-RFP, or MC38, as described in Appendix Supplementary Materials and Methods) according to the experimental needs. In experiments utilizing IFNα/βR$^{-/-}$ mice (inbred C57BL/6) or normal inbred C57BL/6 mice, only C57BL/6-derived MC38 cancer cells were used to avoid immune-mediated rejection of mismatched cells. In selected experiments, CB6 mice were injected intrahepatically (directly beneath the Glisson's capsule) with $5 \times 10^3$ CT26 CRC cells (resuspended in matrigel, BD Bioscience) and 8 days later they were transplanted with Tie2-GFP- or Tie2-IFNα-transduced HSPCs as described in Appendix Supplementary Materials and Methods. Note that this route of injection produced a limited number of hepatic lesions that were confined to the point of injection and to the path of the injecting needle. All mice were maintained in micro-insulator cages under a 12-h light/12-h dark cycle with free access to water and standard mouse diet (Teklad Global 18% Protein Rodent Diet, Harlan) within SPF animal facilities at the San Raffaele Scientific Institute (SRSI). All experiments were carried out in respect of the Permit No. 515 and 691 approved by the SRSI Animal Review Board.

## Depletion of Kupffer Cells (KCs)

Depletion KCs was achieved by intravenous injection of 100 μl clodronate-containing liposomes (Clo-L, from http://www.clodronateliposomes.org) as previously reported (Sitia *et al*, 2011). To obtain higher rates of depletion of KCs, a second administration of 100 μl Clo-L was repeated 3 days after the first one.

## Peripheral blood, intrahepatic leukocytes, and splenocytes analyses

Seven to 10 weeks post-transplant, the whole anti-coagulated blood of Mock/Tie2-GFP and Tie2-IFNα mice was collected from the retro-orbital plexus of anesthetized animals and analyzed as described in Appendix Supplementary Materials and Methods. Intrahepatic leukocytes (IHLs) are isolated from the liver of injected Tie2-GFP mice at specified time points as described in Appendix Supplementary Materials and Methods. The phenotype of circulating WBCs, IHLs, or splenocytes was determined

after red blood cell lysis by flow cytometry analyses, utilizing antibodies summarized in Appendix Supplementary Materials and Methods.

## Immunohistochemistry and Immunofluorescence microscopy

At the time of autopsy, different organs for each mouse were sampled and either fixed in zinc–formalin or 4% paraformaldehyde and processed as described in Appendix Supplementary Materials and Methods. Immunohistochemical staining was performed utilizing the following antibodies: anti-F4/80 (clone A3-1, AbD Serotec); anti-RFP (rabbit polyclonal, ab62341 AbCam); anti-CD34 (clone MEC14.7, Biolegend); anti-Ki67 (clone SP6, Neomarkers); anti-CD3 (clone SP7, AbCam); and anti-CD45R/B220 (clone RA3-6B2, BD Pharmingen). All images were acquired using the Aperio Scanscope CS2 system (Leica Biosystems). Immunofluorescence staining was performed utilizing the following antibodies: anti-GFP (rabbit polyclonal, A11122 Invitrogen) + anti-rabbit Alexa 488 (Invitrogen); anti-MMR (goat polyclonal, AF2535 R&D Systems) + anti-goat Alexa 647 (Invitrogen); anti-F4/80-PE (clone A3-1, AbD Serotec); anti-CD11b-Alexa 647 (clone M1/70; Biolegend); and Hoechst 33342 (Invitrogen). Confocal images were acquired using a Leica TCS SP2 or SP8 confocal system (Leica Microsystems) that are available at the SRSI Advanced Light and Electron Microscopy BioImaging Center (ALEMBIC).

## RNA extraction and quantitative RT–PCR gene expression analyses

Total RNA was isolated from liver homogenates by phenol–chloroform extraction as previously described (Guidotti *et al*, 1995). The extracted RNA was subsequently retro-transcribed to cDNA as previously described (Sitia *et al*, 2011). Quantitative real-time PCR analysis was performed utilizing the 7900HT Fast Real-Time PCR System (Applied Biosystems) as described in Appendix Supplementary Materials and Methods.

## Magnetic resonance imaging (MRI)

CB6 mice, IFN$\alpha/\beta$R$^{-/-}$ mice, and C57BL/6 mice were subjected to *in vivo* abdominal MRI in order to detect liver and peritoneal metastases. Image post-processing was performed using an advanced image segmentation open-source software (Mipav, 5.3.4 version, Biomedical Imaging Research Services Section, ISL, CIT, National Institute of Health, USA) as described in Appendix Supplementary Materials and Methods. All MRI studies were performed at the Preclinical MRI and Ultrasound Facility of the Experimental Imaging Center of SRSI.

## LCMV infection and related procedures

The Armstrong strain of LCMV was utilized in this study (Iannacone *et al*, 2008). Eighty-four or 54 days after receiving saline or CT26 CRC cells, Sham (i.e., mice transplanted with non-transduced HSPCs and intrasplenically injected with saline thereafter) and age-matched CB6 mice (CTRL) or Tie2-IFN$\alpha$ mice were intraperitoneally infected with 200 pfu of LCMV. Whole blood was collected at the indicated

time points from the retro-orbital plexus of anesthetized mice, and white blood cell, hematocrit, and platelet values were measured with an automated cell counter (HeCoVet, Seac-Radim). Single-cell suspensions were prepared from whole blood harvested at day 8 post-infection as previously described (Iannacone *et al*, 2008) and analyzed by flow cytometry as described in Appendix Supplementary Materials and Methods. All infectious work was performed in designated BSL-2 or BSL-3 workspaces, in accordance with SRSI guidelines.

## Characterization of TEMs in human hepatic specimens

Between June and September 2014, patients of the Hepatobiliary Surgery Division at Ospedale San Raffaele scheduled for hepatic resection of colorectal cancer liver metastases were enrolled according to the following criteria: (i) acceptance of the informed consent, (ii) diagnosis of CRC liver metastases or (iii) diagnosis of hepatic hemangioma, and (iv) negativity for HCV and HIV infections. The table shown in Appendix Fig S6A depicts the main clinical and pharmacological characteristics of patients at the moment of enrollment; patients were not matched by number, size, grading of hepatic lesions, or previous pharmacological treatments. No changes in the planned therapeutic regimens were made prior or after this study. The ischemic elapsed time from the hepatic resection was standardized to < 30 min and the sampling was performed by board-certified pathologists on surgical specimens not required for diagnostic purposes. Liver samples were collected from the most distal area of the liver resection (more than 1 cm from the lesion, classified as "distal"), or from the peritumoral area (distance < 1 cm, classified as "proximal"). Non-tumor liver tissue was obtained from a patient that underwent liver resection of a hepatic hemangioma and was used as control. Immediately after sampling, liver specimens were weighted and collected in RPMI medium (Gibco) for intrahepatic leukocyte isolation. The total intrahepatic leukocyte population was isolated and quantified as previously described (de Lalla *et al*, 2004). Single-cell suspensions were subsequently analyzed by flow cytometry with FACS CantoII (BD Pharmingen) and the data processed using FlowJo software (Tree Star Inc.). Human Tie2-expressing monocytes/macrophages (TEMs) were identified as negative surface-stained cells for 7AAD (Biolegend) and positive surface-stained cells for CD45 (clone H130; Biolegend), CD11b (clone M1/70; BD Pharmingen), CD14 (clone M5E2; BD Pharmingen), and Tie2 (clone 83715; R&D systems) as previously described (Venneri *et al*, 2007). Mean fluorescence intensity of TEMs was then normalized subtracting the signal of the matched isotypic antibody control (mouse IgG, clone 11711; R&D systems). The results were expressed as the absolute number of Tie2-positive cells per milligrams of liver tissue.

Formalin-fixed, paraffin-embedded, human livers were cut and stained with hematoxylin/eosin or further processed for immunohistochemical analyses, utilizing anti-TIE2 (goat polyclonal, R&D System) antibody as previously described (Venneri *et al*, 2007); all images were acquired using the Aperio Scanscope CS2 system (Leica Biosystems). The present study was approved by the OSR Ethical Committee (protocol number CE: 33/int 2014) and performed according to the WMA Helsinki declaration.

## Statistical analysis

In all experiments, values are expressed as mean ± SEM. Statistical significance was estimated by two-tailed unpaired parametric Student's *t*-test or Wilcoxon matched pairs test, according to the experimental design as reported in the manuscript. Two-tailed nonparametric Mann–Whitney test was utilized to calculate statistical significance from Gaussian approximations (e.g., to evaluate differences generated as a consequence of tumor growth). The analysis of variance between different time points or recombinant IFNα-treated cell lines of the same experimental group was performed by one-way ANOVA test (nonparametric Kruskal–Wallis variant) and reported in the figure legends. Outlier values were calculated using Grubbs' test (GraphPad Software) and reported in the figure legend. Statistical significance of contingency tables was calculated by two-tailed Fisher's exact test. Statistical significance of survival experiments was calculated by log-rank/Mantel–Cox test. *P*-values < 0.05 were considered statistically significant and reported on graphs. All statistical analyses were performed with Prism 5 (GraphPad Software).

**Expanded View** for this article is available online.

## Acknowledgements

We thank T. Catudella, P. Di Lucia, M. Raso, M. Mainetti, T. Canu, L. Perani, A. Spinelli, D. Covarello, B. Fiore, D. Vignali, and A. Ranghetti for technical assistance in some experiments; M. De Palma, R. Mazzieri, and D. Moi for initial help with the HSPC purification experiments and scientific advice. We thank L. Albarello and C. Doglioni for help with human liver sampling and R. Finazzi for help with patients. We also thank G. Bonizzi, A. De Rose, and G.C. Pruneri (European Institute of Oncology, Milan) for help with IHC staining on human samples. We also thank B. Gentner for critical reading of the manuscript and all the components of the Guidotti and Iannacone laboratories for helpful discussion.

Funding: This work was supported by the following grants: 250219 (LGG) from ERC; GR-2008-1135776, GR-2008-1138756, and RF-2011-02346754 (GS) from Ministero della Salute, The San Raffaele International Postdoctoral Programme PCOFUND-GA-2010-267264 INVEST (MC).

## Author contributions

MC designed and performed research, analyzed data, and wrote the manuscript; AM performed quantitative real-time PCR experiments, human TEMs characterization, and provided intellectual input; GE performed lentiviral particles production, transduction, and VCN assessment; AF performed immunohistochemistry experiments; NLT performed research and MTT assays; RA performed experiments with LCMV; PM and AE performed MRI analyses; FC and LA performed research; MI provided intellectual input and scientific advice; LN and LGG provided intellectual input and scientific advice and edited the manuscript; GS performed and supervised research, analyzed data, and wrote the manuscript.

## Conflict of interest

LN is an inventor on patents describing microRNA-regulated LVs owned by San Raffaele Scientific Institute and Telethon Foundation; LN is also a founder and owns equity in Genenta Science, a biotechnology startup aiming to develop targeted delivery of IFNα to hematopoietic tumors. The remaining authors declare that they have no conflict of interest.

### The paper explained

#### Problem

Colorectal cancer (CRC) is one of the most common malignancies in humans and one of the leading causes of cancer-related deaths worldwide. Most of these deaths relate to the presence and progression of liver CRC metastases. In order to identify a novel and more effective therapy for this disease, we adopted a gene transfer strategy into mouse hematopoietic stem/progenitor cells to generate immune-competent mice in which TEMs—a Tie2[+] subset of monocytes/macrophages found at peritumoral sites—express interferon-alpha (IFNα), a pleiotropic cytokine with anti-tumor effects, to verify the impact of such approach in mouse models of CRC liver metastases.

#### Results

We have shown that targeted delivery of IFNα by gene/cell therapy to the liver safely prevents, reduces, or reverts the growth of hepatic CRC metastases improving overall survival of immune-competent mice, without causing systemic side effects, hematopoietic toxicity, or inability to respond to a virus challenge. This study also identified the liver microenvironment as a crucial target of TEM-mediated IFNα anti-tumor effect.

#### Impact

These results indicate that targeted IFNα delivery to the liver could be adopted as additional adjuvant therapy in patients with CRC liver metastases.

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
