## [Review Process File · EMBO Molecular Medicine]

IFN gene/cell therapy curbs colorectal cancer colonization of the liver by acting on the hepatic microenvironment

Mario Catarinella, Andrea Monestiroli, Giulia Escobar, Amleto Fiocchi, Ngoc Lan Tran, Roberto Aiolfi, Paolo Marra, Antonio Esposito, Federica Cipriani, Luca Aldrighetti, Matteo Iannacone, Luigi Naldini, Luca G. Guidotti and Giovanni Sitia

Corresponding author: Giovanni Sitia, IRCCS San Raffaele Scientific Institute

Review timeline:

Submission date:	29 April 2015
Editorial Decision:	16 June 2015
Revision received:	04 November 2015
Editorial Decision:	18 November 2015
Revision received:	25 November 2015
Accepted:	04 December 2015

Transaction Report:

Editor: Roberto Buccione

1st Editorial Decision

16 June 2015

Thank you for the submission of your manuscript to EMBO Molecular Medicine. We are sorry that it has taken longer than usual to get back to you on your manuscript. In this case we experienced some difficulties in securing three appropriate expert reviewers and then obtaining their evaluations in a timely manner and furthermore, we needed to discuss your manuscript further.

As you will see the three Reviewers are globally positive, but do raise many issues, one of which is especially critical and shared by all three. Although I will not dwell into much detail, I would like to highlight this main point.

You will see that the main recurring theme with a clear consensus is the unconvincing "metastasis" setting used in the manuscript. For instance, Reviewer 2 goes into much detail to discuss the issue of clinical relevance. Indeed s/he notes, and we agree, that the main merit and value of the manuscript is to illustrate a potential direct clinical application. For this reason, it becomes imperative to confirm and validate the main conclusions in an appropriate bona fide metastasis model. The reviewers offer various suggestions on how to achieve this goal. I should add that we fully agree that this is a current limitation of the manuscript but also that to ameliorate this aspect would significantly increase the impact, appeal and reach of your work.

Reviewer 3 would also like you to develop the mechanistic aspect a little more and suggests some approaches to that effect.

The Reviewers point to other issues, which while important and requiring your action, appear less

demanding and difficult to tackle.

In conclusion, while publication of the paper cannot be considered at this stage, given the potential interest of your findings and after internal discussion, we have decided to give you the opportunity to address the above concerns. We are thus prepared to consider a substantially revised submission, with the understanding that the Reviewers' concerns must be addressed in toto, with additional in vivo experimentation as appropriate and that acceptance of the manuscript will entail a second round of review.

I appreciate that if you do not have the required data available at least in part, to address the above, this might entail a significant amount of time, additional work and might be technically challenging, I would therefore understand if you chose to rather seek publication elsewhere at this stage, which of course we hope that you will not. Should you do so, we would welcome a message to this effect.

Please note that it is EMBO Molecular Medicine policy to allow a single round of revision only and that, therefore, acceptance or rejection of the manuscript will depend on the completeness of your responses included in the next, final version of the manuscript.

As you know, EMBO Molecular Medicine has a "scooping protection" policy, whereby similar findings that are published by others during review or revision are not a criterion for rejection. However, I do ask you to get in touch with us after three months if you have not completed your revision, to update us on the status. Please also contact us as soon as possible if similar work is published elsewhere.

EMBO Molecular Medicine now requires a complete author checklist (<http://embomolmed.embopress.org/authorguide#editorial3>) to be submitted with all revised manuscripts. Provision of the author checklist is mandatory at revision stage; The checklist is designed to enhance and standardize reporting of key information in research papers and to support reanalysis and repetition of experiments by the community. The list covers key information for figure panels and captions and focuses on statistics, the reporting of reagents, animal models and human subject-derived data, as well as guidance to optimise data accessibility.

I look forward to seeing a revised form of your manuscript in due time.

***** Reviewer's comments *****

Referee #1 (Comments on Novelty/Model System):

In the manuscript "IFN gene/cell therapy restricts colorectal cancer colonization of the liver primarily acting on the hepatic microenvironment", Catarinella et al. describe the anti-tumorigenic effect of IFN α delivered by Tie2⁺ monocytes/macrophages (TEMs) on liver colonizing colorectal cancer cells CT26 and MC38, which were introduced via intrasplenic injection. The major issue with this manuscript lies in the interpretation of the CT26 liver metastasis dataset (and presenting it as its main focus), according to which IFN α delivered by TEMs effectively "inhibits tumor growth and improves survival". The dramatic difference in tumor incidence, tumor volume, and tumor burden, between the Tie2-GFP group and Tie2-IFN α group at the various time points (as shown in Figure 2 and 3) could be mainly attributed to the failure of CT26 cells to graft/colonize the liver after their introduction in the presence of IFN α . Initial cell engraftment/colonization needs to be measured. The MC38 liver metastasis dataset (Fig. S5) suggests that with better initial colonization of CRC cells, IFN α only delays tumor development and mouse death. To examine the real potential of the Tie2-IFN α for the treatment of human CRC liver metastases, the authors should exclude this "failure to colonize" possibility by introducing Tie2-IFN α /GFP after the transplanted CRC cells have already established liver colonies, i.e. "metastatic CRC cells growing in the liver".

Minor issues:

Fig 2D. The CD8⁺/IFN γ ⁺ cell population appears significantly different between the Sham and the Tie2-IFN α groups at 8 days post-LCMV infection, which contradicts the authors' statement "HSPC transplantation targeting TEM-mediated IFN α did not alter the host capability to respond to virus

challenge".

Fig 4. The authors should include the NaCl group dataset. Also, the authors should perform a similar study to the one in Fig 4 to analyze MC38 liver colonies in the presence of Tie2-GFP/IFN α .

Fig 5. The authors focused on the MC38 liver metastasis model here to address the targets of IFN α . There is no mention of a parallel experiment done with the CT26 liver metastasis model. Did the authors obtain a similar conclusion with CT26 cells? Are CT26 cells more sensitive to direct IFN α exposure than MC38?

Fig 5 and S7. The increased vector copy number in Tie2-GFP derived from IFN α /bR KO bone marrow seems to have anti-tumor properties compared to Tie2-GFP derived from B6 bone marrow. Similarly, will the Tie2-IFN α derived from IFN α /bR KO bone marrow become even better anti-tumor agent?

Fig 5S. Tumor volume between d54 Tie2-IFN α group and d16 Tie2-GFP group seems similar, yet several of these d54 Tie2-IFN α tumor-bearing mice needed to be euthanized. Can the authors explain the death of these mice?

Referee #2 (Comments on Novelty/Model System):

See comments to the author for further details

Referee #2 (Remarks):

This paper shows that mice bearing bone marrow transplants of HSCs engineered *ex vivo* to express Tie2-targeted delivery of IFN α , exhibit decreased progression of hepatic tumors in an intrasplenic seeding assay. As such it can be seen as being of limited novelty as it confirms the previous studies by its authors in different tumor models (De Palma et al. 2008; Escobar et al. 2014). However, it is not without merit as it provides the first pre-clinical evidence for this therapeutic approach having possible utility in the treatment of liver metastasis in CRC patients. It is this possible clinical application which makes the paper publication-worthy. However, a number of issues/points need to be addressed before it can be considered for this. These are as follows:

1. The 'metastasis' assay used throughout is really a seeding assay - ie. one in which CRC cell lines are injected into the spleen and then make their way into the liver. The clinical relevance of this model needs to be explained and justified in the paper, and the major findings of this study confirmed in a transgenic/spontaneous model of CRC metastasis (ie. in which HSCs are transplanted several weeks before the liver metastases form).
2. Also, as pointed out by the authors in the Discussion, this HSC therapy is most likely to be adopted as an 'adjuvant therapy in patients at high risk of developing metastasis' - so they should attempt to show its efficacy in a clinically-relevant adjuvant model of CRC metastasis (or explain in the Discussion why this is not possible). Moreover, the authors need to show that their genetically engineered HSCs can repopulate the bone marrow of mice with CRC (as the patients likely to be enrolled in clinical trials of this therapy will already have CRC).
3. The authors should also discuss the possible confounding side effects of this HSC therapy in patients who develop diseases (after HSC transplantation) in which TEMs are upregulated (or M2-activated macrophages - as Tie2 is a marker of these cells) (Krausz et al. 2012; Patel et al. 2013; Mantovani et al. 2013).
4. Why do few HSC-derived, Tie2-GFP TEMs in mice not make their way into liver metastases - rather sit in 'proximal' locations in the liver? How can the authors exclude the possibility that TEMs have reduced viability in CRC metastases? According to Fig. 6, TEMs also take up this position naturally in the livers of CRC patients so it is clearly not due to their IFN expression in mice. Finally, immunostaining of these liver sections should be provided so the reader can directly compare TEM location in mice (Fig.1) and human (Fig. 6) livers.
5. Page 16 mentions Fig. 1C, which is missing.
6. IFN α by TEMs expression should be analysed in a range of healthy tissues in their tumor-bearing mice (eg. by immunostaining of tissue sections) to show off-target expression is minimal.
7. Discrepancies in the effects of TEM delivery of IFN on immune cell activation in this study and their previous one in a different syngeneic tumour model (Escobar et al. 2014) are inadequately addressed on page 19, and should be investigated in further experiments and included in the revised

paper.

Referee #3 (Comments on Novelty/Model System):

The level of development of the cell therapy/gene transfer is very good, and of potentially high medical impact. However, the mouse model has two problems: mouse CRC cell lines (CT26 and MC38) are not known to produce very human-like tumours (CT26 appears mesenchymal, not epithelial), but there is no better alternative at the moment.

More importantly then, the setting the models are used in (prevention of liver colonization) neither represents the treatment of established metastases, nor faithfully approximates the adjuvant setting (see point 1 in the review for suggestions of improvement).

Referee #3 (Remarks):

The authors use a previously described HSPC gene transfer strategy to deliver IFN γ to the liver via Tie2⁺ monocytes/macrophages and inhibit CRC metastasis from murine cell lines, injected intrasplenically. This pre-clinical study is a continuation of the group's existing body of work, and aims to provide the rationale for human trials. Although the observations are very interesting and timely, there are several major points of criticism.

1)As the authors rightfully discuss in the ms, their model of performing cell/gene therapy on naïve mice well before the injection metastatic cells has very little to do with the treatment of established (refractory) metastases, which is also the most common setting for the testing of novel therapies. Indeed, the mouse model used is somewhat more faithful to the adjuvant setting, which the authors propose is a good option for their treatment strategy. Adjuvant trials are performed after curative resection and extensive chemotherapy, neither of which are tested in the current study. To strengthen the rationale for adjuvant trials, the authors should do one or both of the following experiments: a) inject CRC cell lines in the colon/caecum before cell/gene therapy and score metastases with IFN γ or control TEMs; b) same injection as in (a) but treating with chemo and/or resecting primary tumours before starting the cell/gene therapy.

2)The mechanism of action is not well developed. In Figure 4, the authors quantify cell types in a single time point where the tumour burden differs ~50-fold (Figure 3C, % RFP; one of the conditions has hardly any tumour cells), making the comparison very skewed. Thus, it is unclear what the analysis contributes to a better understanding. In Figure 5, it is necessary to show that the WT reconstituted BM in irradiated IFN α /bR^{-/-} animals is successful and fully functional, in order to conclude that no BM-derived cell is responsible for the IFN γ -mediated tumour rejection.

3)As the data may indicate that the effect of IFN γ is not expected to require Kupffer cells, BM-derived cells or to be occurring within the cancer cells themselves, it remains unclear where it is signaling and to what end. The authors should do IHC/immune fluorescence stainings for downstream signaling events (e.g. pSTAT1) to help identify the cell types that may be responsible for the observed effects.

1st Revision - authors' response

04 November 2015

Thank you for your letter regarding our manuscript **"IFN α gene/cell therapy restricts colorectal cancer colonization of the liver acting primarily on the hepatic microenvironment"**. We also thank the referees for their reviews and helpful comments. Based on their constructive comments we have extensively revised our manuscript and added new substantial experimental data, which we believe not only significantly enhance the manuscript but also sufficiently address the referees' major and minor concerns.

In summary, we tried several strategies to address the major questions raised by all three referees. Indeed it would be great to have immunocompetent mouse models that

recapitulate different aspects of CRC progression in humans, for instance, mice in which we can monitor the relative growth of liver metastases that arise after chemotherapy or surgical resection of primary tumors in the colon or, alternatively, liver metastases that form after direct injection of CRC cells in the large intestine. Unfortunately, mouse models of spontaneous CRC capable of producing liver metastases do not exist and the injection of CRC cells in the cecal wall followed by bone marrow (BM) transplantation turned out to be not really practical. In fact, we injected the two CRC cell lines into the cecal wall of immunocompetent CB6 mice, as previously reported (Bresalier et al, 1987), and we succeeded at promoting the development of orthotopic CRC in about 60% of the mice. In agreement with previous publications (Bresalier et al, 1987; Zhang et al, 2013), however, by the time (i.e. ~45 days after injection) that we were forced to euthanize the animals for ethical reasons (note that the intestinal tumors became large and the general health of the animals rapidly deteriorated), none of the tumor-bearing mice showed liver metastatic foci. Furthermore, treating established CRC liver metastases, turned out to be also challenging because the irradiation procedure, necessary for BM transplantation could by itself impact the growth rate of intrahepatic established CRC metastases, that we had to take into account.

For the abovementioned reasons, we chose to inject wild-type mice with a small number of CRC cells (5×10^3 CT26-RFP) directly beneath the Glisson's capsule in a localized area of the left hepatic lobe (Fig 5A and C of the revised version). Eight days after injection, the mice were divided in two groups and subjected to BM transplantation with HSPCs previously transduced to produce either Tie2-GFP or Tie2-IFN α cells. Four days later (a time point in which Tie2-GFP or Tie2-IFN α cells are not yet emerged from the BM (Lechman et al, 2012; Zonari et al, 2013)), the animals were subjected to the first MRI analysis, which - as expected - detected no significant differences in the total tumor volume from the liver of the two groups of mice. Note that at this time point post-transplant mice could be safely transferred to the MRI facility in compliance with our biosafety protocols. Follow-up MRI analyses at days 21 and 28 post-transplant revealed that Tie2-IFN α mice display reduced volumes of CRC liver metastases when compared to those detected in Tie2-GFP mice (Fig 5B and C of the revised version). This difference fell short of being statistically significant, probably because of the partial hematopoietic reconstitution at these time points (note that at day 28 after BM transplantation, peripheral WBCs were still about 30% below normal counts, Appendix Fig S5A). Remarkably, Tie2-IFN α mice showed a statistically significant increase - over Tie2-GFP mice - of *Irf7*, an IFN α -stimulated gene at this early follow up time (Fig 5F) and a statistically significant improvement in overall survival at later time (Fig 5H of the revised version). These results were accompanied by a statistically significant reduction in the % of Tie2-IFN α mice that develop peritoneal lesions (a complication of Glisson's capsule infiltration with CRC cells spreading into the peritoneal cavity). Indeed, less than 30% of Tie2-IFN α mice showed MRI-based evidence of peritoneal carcinomatosis at day 28, while 100% of Tie2-GFP mice developed such complication at this same time point (Fig 5D and Movie EV8 of the revised version).

All in all, the results abovementioned are of particular importance because they positively address a critical question raised by all the 3 referees, i.e. to evaluate the antitumor potential of our approach on CRC liver metastases that are already established. While suggesting that the "Tie2-IFN α approach" could be employed as adjuvant therapy treating established (e.g. visible) metastases, we would like to point out that these results also suggest that such procedure may also be considered to prevent/contain the growth of metastases that are too small to be diagnosed at the moment of surgical resection.

To aid the referees in the re-evaluation of the study, all relevant passages in the revised version have been underlined.

Response to Referee #1:

The major issue with this manuscript lies in the interpretation of the CT26 liver metastasis dataset (and presenting it as its main focus), according to which IFN α delivered by TEMs effectively "inhibits tumor growth and improves survival". The dramatic difference in tumor

incidence, tumor volume, and tumor burden, between the Tie2-GFP group and Tie2-IFN α group at the various time points (as shown in Figure 2 and 3) could be mainly attributed to the failure of CT26 cells to graft/colonize the liver after their introduction in the presence of IFN α . Initial cell engraftment/colonization needs to be measured.

We thank the referee for raising this issue, which - apparently - was not properly explained. Indeed, we actually quantified the early phases of tumor cell engraftment and liver colonization in the original version: as described in the original Fig 3B, C and Supplementary Fig S6A (Fig 3B and C and Appendix Fig S3A of the revised version), we did so by: i) intrasplenically injecting a relatively high dose of colorectal cancer (CRC) cells (5×10^5 CT26-RFP cells/mouse) into either Tie2-GFP mice or Tie2-IFN α mice, ii) sacrificing the mice 5 min thereafter and iii) analyzing livers for the relative content of CT26-RFP cells through RFP-specific molecular and histopathological assays (real-time PCR and immunohistochemistry). Indeed, both assays showed that similar numbers of CT26-RFP cells reach the liver of Tie2-GFP and Tie2-IFN α mice at 5 min post-injection, indicating that the slightly higher-than-normal levels of IFN α detected prior to CRC cell injection in the former mice (Fig 3A of the original and revised version) does not affect the capacity of CRC cells to initially engraft the liver parenchyma. Moreover, these cells not only arrest similarly in Tie2-GFP and Tie2-IFN α mice but they remain viable on site as we could detect them at follow up time points (Fig 3B), yet they failed to proliferate as in controls. Indeed, in some mice we did observe MRI-detectable liver lesions, which then regressed (Fig 2), indicating that the metastatic growth of CRC cells is impaired only at later time points, as denoted by the significant difference in tumor volumes at days 3 and 7 post-injection between the two groups of mice (Fig 3B and C of the original and revised version). Note that it is not possible to quantify the early (i.e. 5 min post-injection) hepatic accumulation of CT26-RFP cells in these animal models when a dose of CRC cells (5×10^3 CT26-RFP cells/mouse) identical to the low one described in Fig 1 and 2 is used (data not shown), as the relative amount of RFP-derived gene products is below the detection limit of our assays.

The MC38 liver metastasis dataset (Fig. S5) suggests that with better initial colonization of CRC cells, IFN α only delays tumor development and mouse death. To examine the real potential of the Tie2-IFN α for the treatment of human CRC liver metastases, the authors should exclude this "failure to colonize" possibility by introducing Tie2-IFN α /GFP after the transplanted CRC cells have already established liver colonies, i.e. "metastatic CRC cells growing in the liver".

We fully agree with the referee's concern and as abovementioned, followed his/her suggestion of examining the effect of Tie2-IFN α on established liver metastases. To this end, 15 mice were injected with a small number of CRC cells (5×10^3 CT26-RFP) directly beneath the Glisson's capsule in a localized area of the left hepatic lobe (Fig 5A and C of the revised version). Eight days after injection, the mice were divided in two groups and subjected to bone marrow (BM) transplantation with HSPCs previously transduced to produce either Tie2-GFP or Tie2-IFN α cells. Four days later (a time point in which Tie2-GFP or Tie2-IFN α cells are not yet emerged from the BM (Lechman et al, 2012; Zonari et al, 2013), the animals were subjected to the first MRI analysis, which - as expected - detected no significant differences in the total tumor volume from the liver of the two groups of mice. Note that at this time point post transplant mice could be safely transferred to the MRI facility in compliance with our biosafety protocols. Follow-up MRI analyses at days 21 and 28 post-transplant revealed that Tie2-IFN α mice display reduced volumes of CRC liver metastases when compared to those detected in Tie2-GFP mice (Fig 5B and C of the revised version). This difference fell short of being statistically significant, probably because of the partial BM reconstitution at these time points (note that at day 28 after BM transplantation, peripheral WBCs were still about 30% below normal counts, Appendix Fig S5A). Remarkably, Tie2-IFN α mice showed a statistically significant increase - over Tie2-GFP mice - of hepatic Irf7 (Fig 5F) and a statistically significant improvement in overall survival (Fig 5H of the revised version). These results were accompanied by a statistically significant reduction in the % of Tie2-IFN α mice that develop peritoneal lesions (a complication of Glisson's capsule infiltration with CRC cells spreading into the peritoneal cavity). Indeed, less than 30% of Tie2-IFN α mice showed MRI-based evidence of

peritoneal carcinomatosis at day 28, while 100% of Tie2-GFP mice developed such complication at this same time point (Fig 5D and Movie EV8 of the revised version). Text regarding these results has been added in the revised manuscript at pages 15-17.

Minor issues:

Fig 2D. The CD8⁺/IFN γ ⁺ cell population appears significantly different between the Sham and the Tie2-IFN α groups at 8 days post-LCMV infection, which contradicts the authors' statement "HSPC transplantation targeting TEM-mediated IFN α did not alter the host capability to respond to virus challenge".

While, there was no statistical difference between the two small groups of animals depicted in the original Fig 2D (only one Mock transduced mouse showed a % of circulating CD8⁺/IFN γ ⁺ cells that was higher than in all other animals), we repeated the LCMV infection with some additional animals from these groups (where LCMV infection occurs 54 days after NaCl or CRC cell injection in Sham/CTRL or Tie2-IFN α mice, respectively). Results emerging from these additional experiments (see the revised Fig 2F and Appendix Fig S2) strengthen the notion that the % of circulating CD8⁺/NP118⁺ T cells capable of producing IFN γ is comparable between Sham/CTRL and Tie2-IFN α mice, suggesting that HSPC transplantation with Tie2-IFN α LVs does not significantly alter virus-specific CD8⁺ T cell responses. Text regarding these results has been added in the revised manuscript at page 8 line 20 and Appendix page 23.

Fig 4. The authors should include the NaCl group dataset. Also, the authors should perform a similar study to the one in Fig 4 to analyze MC38 liver colonies in the presence of Tie2-GFP/IFN α .

We agree with the referee's comment and added the NaCl group dataset as additional panels in Fig EV4 of the revised version as well as additional bars in the corresponding quantifications.

As per the second comment, we respectfully disagree with the idea of performing more experiments with the MC38 cell line. In our opinion - as biased as an opinion from an author might be - little or no additional mechanistic insight will emerge from such experiments and, therefore, the amount of resources necessary to perform those experiments is not clearly justified.

Fig 5. The authors focused on the MC38 liver metastasis model here to address the targets of IFN α . There is no mention of a parallel experiment done with the CT26 liver metastasis model. Did the authors obtain a similar conclusion with CT26 cells? Are CT26 cells more sensitive to direct IFN α exposure than MC38?

Due to MHC mismatch - which by itself may lead to tumor cell rejection - it is technically not possible to reproduce the experiment described in the original Fig 5 (now revised Fig 4) with CT26 cells (BALBc-derived, H-2^d restricted). Indeed we injected MC38 cells (C57BL/6-derived, H-2^b restricted) into the experimental setting of IFN α / β R^{-/-} mice (inbred C57BL/6, H-2^b restricted) thus maintaining a syngeneic experimental design. Moving the IFN α / β R^{-/-} alleles into a BALBc background would be necessary to perform the suggested experiments but that would require 2-3 years (>10 generations) of mouse backcrossing.

As per the relative sensitivity of the two CRC cell lines towards IFN α , we have quantified the extent to which CT26 or MC38 CRC cells show a reduced proliferation *in vitro* upon exposure to recombinant IFN α . As shown in the revised Fig EV1A, both cell lines are rather sensitive to the anti-proliferative effect of IFN α , with a tendency of MC38 cells of being more sensitive (especially at the highest dose). This indicates that - at least *in vitro* - CT26 cells appear to be less sensitive to IFN α than MC38 cells. Text regarding these results has been added in the revised manuscript at page 6, lines 5-6.

Fig 5 and S7. The increased vector copy number in Tie2-GFP derived from IFN α /bR KO

bone marrow seems to have anti-tumor properties compared to Tie2-GFP derived from B6 bone marrow. Similarly, will the Tie2-IFN α derived from IFN α / β R KO bone marrow become even better anti-tumor agent?

The referee suggests a potential link between vector copy number (VCN) and antitumor potential. To shed some light on this, we first performed an *in vitro* titration of the Tie2-IFN α LV in IFN α / β R^{-/-} HSPCs. Reducing the amount of lentiviral transducing units (TU/ml) of almost 20 fold (from 50x10⁶ TU/ml to 2.65x10⁶ TU/ml) resulted in a significant ~2-fold decrease in VCN (from a VCN of ~8 to a VCN of ~4), see Figure below. [Figure omitted upon request by the authors] Of note, treating wild type (IFN α / β R^{+/+}) HSPCs with 66x10⁶ TU/ml resulted in a VCN of ~5.

We then performed a new set of experiments in which C57BL/6 mice were reconstituted with IFN α / β R^{-/-} BM transduced with either Tie2-GFP or Tie2-IFN α LVs at a TU/ml dose of 2.5x10⁶. This TU/ml dose is 40 fold lower than that used throughout the original manuscript and resulted in a 2 fold lower VCN (see Panel A of Appendix Fig S4 of the revised version). The notion that the (reduced) tumor growth observed in these animals (see Fig 4 of the revised version) was very similar to that observed in similarly treated animals bearing Tie2-IFN α cells with a 2 fold higher VCN (see Figure S7 from the original version) suggests that VCN is not linked to antitumor efficiency. Furthermore, new analyses of CRC metastatic growth at days 14 or 21 post-transplantation in C57BL/6 mice reconstituted with either IFN α / β R^{-/-} Tie2-GFP HSPCs (VCN of ~6) or C57BL/6 Tie2-GFP HSPCs (VCN of ~2) or in IFN α / β R^{-/-} mice reconstituted with C57BL/6 Tie2-GFP HSPCs (VCN of ~1) (see Appendix Fig S4 of the revised version) revealed that the tumor volume was somewhat smaller in the former group at the earliest but not at the latest time point (where the volume was actually bigger). Since these latest experiments utilizing lower VCN better convey our message, we now show these results in new Fig 4. Text regarding these results has been added in the revised manuscript at page 13 lines 2-5.

Fig 5S. Tumor volume between d54 Tie2-IFN α group and d16 Tie2-GFP group seems similar, yet several of these d54 Tie2-IFN α tumor-bearing mice needed to be euthanized. Can the authors explain the death of these mice?

We thank the referee for having raised this issue, which was not properly presented in the original version. Panel A of Fig EV3 from the revised version depicts the percentage of tumor bearing mice of the two experimental groups (Tie2-GFP or Tie2-IFN α mice). Two out of the 5 Tie2-IFN α mice developed MRI-detectable liver tumors by day 16-20, with an additional animal that displayed at least one liver lesion by day 30 (this is in comparison with the 100 % positivity detected in Tie2-GFP mice by day 16 already). Of the 3 Tie2-IFN α mice with liver metastases, the one depicted in Fig EV3B by a blue hexagon had to be euthanized at day 44 because of tumor growth. A second animal (depicted by a blue triangle) was subjected to liver MRI on day 54 and found dead on the following day. As we were unable to define the exact cause of this death, we did not censor this event. We apologize for the misunderstanding and corrected Fig EV3 panel A. We also added new text describing this issue in the legend to Fig EV3 (pages 48-49).

Response to Referee #2:

1. The 'metastasis' assay used throughout is really a seeding assay - ie. one in which CRC cell lines are injected into the spleen and then make their way into the liver. The clinical relevance of this model needs to be explained and justified in the paper, and the major findings of this study confirmed in a transgenic/spontaneous model of CRC metastasis (ie. in which HSCs are transplanted several weeks before the liver metastases form).

We thank the referee for having raised these issues. Indeed, it would be great to have immunocompetent mouse models where to verify the benefit of our Tie2-IFN α approach (for instance, mice in which we can monitor the relative growth of liver metastases that arise after surgical resection of primary tumors in the colon). Unfortunately, these models

do not exist. Mouse models of intestinal adenomas where cancer can advance to the locally invasive stage have been developed, but none of these models are characterized by liver metastasis (Kobaek-Larsen et al, 2000; Taketo & Edelmann, 2009). That the Tie2-IFN α approach positively impact metastases arising from spontaneous primary tumors has been suggested in previous publications utilizing the PyMT breast cancer model (De Palma et al, 2008; Escobar et al, 2014). Here, we developed a model specifically designed to interrogate the impact of the Tie2-IFN α approach on the growth of CRC cells that have reached the liver. Text has been added in the revised manuscript at page 6, lines 7-9.

2. Also, as pointed out by the authors in the Discussion, this HSC therapy is most likely to be adopted as an 'adjuvant therapy in patients at high risk of developing metastasis' - so they should attempt to show its efficacy in a clinically-relevant adjuvant model of CRC metastasis (or explain in the Discussion why this is not possible). Moreover, the authors need to show that their genetically engineered HSCs can repopulate the bone marrow of mice with CRC (as the patients likely to be enrolled in clinical trials of this therapy will already have CRC).

We thank the referee for having raised this important issue and, as already described above in the rebuttal to referee #1, we followed his/her suggestion of examining the effect of Tie2-IFN α on established liver metastases. To this end, we injected a small number of CRC cells (5×10^3 CT26-RFP) in a specific area of the left hepatic lobe (directly beneath the Glisson's capsule) of C57BL/6 mice ($n=15$) (Fig 5A and C of the revised version). Eight days after injection, the mice were divided in two groups and subjected to whole body irradiation and bone marrow (BM) transplantation with HSPCs previously transduced to produce either Tie2-GFP or Tie2-IFN α cells. Four days later (a time point in which Tie2-GFP or Tie2-IFN α cells are not yet emerged from the BM (Lechman et al, 2012; Zonari et al, 2013), the animals were subjected to the first MRI analysis, which - as expected - detected no significant differences in the total tumor volume from the liver of the two groups of mice. Note that at this time point post transplant mice could be safely transferred to the MRI facility in compliance with our biosafety protocols. Follow-up MRI analyses at days 21 and 28 post-transplant revealed that Tie2-IFN α mice display reduced volumes of CRC liver metastases when compared to those detected in Tie2-GFP mice (Fig 5B and C of the revised version). This difference fell short of being statistically significant, however, probably because of the partial BM reconstitution at these time points (note that at day 28 after BM transplantation, peripheral WBCs were still about 30% below normal counts, Appendix Fig S5A). Remarkably, Tie2-IFN α mice showed a statistically significant increase - over Tie2-GFP mice - of liver *Irf7* - a prototypical IFN-stimulated gene (Fig 5F of the revised version) - and a statistically significant improvement in overall survival (Fig 5H of the revised version). These results were accompanied by a statistically significant reduction in the % of Tie2-IFN α mice that develop peritoneal lesions (a complication of Glisson's capsule infiltration with CRC cells spreading into the peritoneal cavity). Indeed, less than 30% of Tie2-IFN α mice showed MRI-based evidence of peritoneal carcinomatosis at day 28, while 100% of Tie2-GFP mice developed such complication at this same time point (Fig 5D and Movie EV8 of the revised version). Text regarding these results has been added in the revised manuscript at pages 15-17.

As per the request of showing that genetically engineered HSPCs can repopulate the BM of mice bearing CRC, we'd like to point out that the experiments above-mentioned (where CRC cell injection preceded HSPC transplantation by 8 days) indicated that the extent of BM repopulation (monitored by looking at complete blood counts (CBC) at day 28 post-transplantation, Appendix Fig S5A) was similar between Tie2-GFP mice and Tie2-IFN α mice in which CRC metastases grow at a different speed. This suggests that preexisting CRC liver metastases seem not to impact the capacity of genetically engineered HSPCs to repopulate the bone marrow. Text regarding these results has been added in the revised manuscript at pages 15-17.

3. The authors should also discuss the possible confounding side effects of this HSC therapy in patients who develop diseases (after HSC transplantation) in which TEMs are upregulated (or M2-activated macrophages - as Tie2 is a marker of these cells) (Krausz et

al. 2012; Patel et al. 2013; Mantovani et al. 2013).

This is an important aspect in the preclinical evaluation of our approach necessary before moving to the clinic. Indeed, we monitored some possible side effects in our models, and didn't identify any apparent dysfunction or pathology (at time of autopsy) of the BM (monitored by CBC in blood) or other organs (Fig EV2B of the revised version) as a consequence of our approach. Moreover, considering that TEMs have been described to play a role in wound healing (De Palma et al, 2008) and that CRC cell injection in our mouse model requires the incision/suture of skin and peritoneum, it is noteworthy that no differences in the wound healing process (time of repair) was observed among Tie2-GFP mice, Tie2-IFN α mice or Sham controls, similarly to what previously reported (De Palma et al, 2008). Nevertheless, some of these additional concerns, will have to be tested in specific pre-clinical animal models, in order to exclude these additional side effects. Moreover, additional safety strategies, such as the genetic modification of only a fraction of HSPCs to limit potential IFN α exposure and toxicity will have to be developed before clinical translation. Note that Tie2-IFN α transduced HSPCs may eventually exhaust over time spontaneously as previously reported (Escobar et al, 2014), thus limiting the long term exposure to this cytokine.

4. Why do few HSC-derived, Tie2-GFP TEMs in mice not make their way into liver metastases -rather sit in 'proximal' locations in the liver? How can the authors exclude the possibility that TEMs have reduced viability in CRC metastases? According to Fig. 6, TEMs also take up this position naturally in the livers of CRC patients so it is clearly not due to their IFN α expression in mice. Finally, immunostaining of these liver sections should be provided so the reader can directly compare TEM location in mice (Fig.1) and human (Fig. 6) livers.

We thank the referee for having raised these important issues. New data depicted in Fig 5E of the revised version indeed strengthens the concept that TEMs are preferentially positioned in peritumoral areas, consistently with previous publications and their role as pro-angiogenic factors (De Palma et al, 2008; De Palma et al, 2005; Venneri et al, 2007). However, we can't exclude the possibility that TEMs enter CRC lesions and eventually die in such environment. The peritumoral localization of TEMs seems also apparent in new experiments we performed on human liver sections stained for TIE2 (Fig 6 of the revised version). To this end, we carried out numerous attempts to identify TIE2 positive cells by immunostaining either on frozen or on formalin fixed liver tissue. We used 4 different commercially available antibodies sold to be specific for human TIE2, and only one of them (a polyclonal goat anti-TIE2 from R&D Systems) turned out to produce a specific staining (Fig 6B and C). The notion that sites that are distal from the lesion (more than 1 cm from it) contain only TIE2⁺ cells with an apparent endothelial morphology (characterized by an elongated appearance) while sites that are close to the lesion (less than 100 μ m from it) also contain TIE2⁺ cells with an apparent monocyte-like morphology (characterized by a round appearance) suggests a peritumoral localization of human TEMs as well. Text regarding these results has been added in the revised manuscript at pages 17 line 23 and page 18 lines 1-6.

5. Page 16 mentions Fig. 6C, which is missing.

We thank the referee and corrected the mistake.

6. IFN α by TEMs expression should be analysed in a range of healthy tissues in their tumor-bearing mice (eg.by immunostaining of tissue sections) to show off-target expression is minimal.

We thank the referee for this comment and performed new molecular assays looking at interferon stimulated genes in non-tumor brains and non-tumor kidneys of liver metastasis-bearing Tie2-GFP or Tie2-IFN α mice (Fig 2D and E of the revised version). We found no significant differences between the two groups of mice in interferon induced genes at these sites. Text regarding these results has been added in the revised manuscript at pages 7

line 23 and 8 lines 1-3.

As also suggested by referee #3, we performed immunohistochemical analyses for the phosphorylated form of signal transducer and activator of transcription 1 (pSTAT1, a IFN α downstream signaling molecule) on livers and other “non-tumor tissues” from Tie2-IFN α mice or Sham mice at the latest time point available (Day 450) post-transplantation. As shown below, pSTAT1 staining of different organs did not differ significantly between the two groups of mice, suggesting once more that in our setting the off-target IFN α expression by TEMs is minimal.

7. Discrepancies in the effects of TEM delivery of IFN α on immune cell activation in this study and their previous one in a different syngeneic tumour model (Escobar et al. 2014) are inadequately addressed on page 19, and should be investigated in further experiments and included in the revised paper.

We thank the referee for having raised this issue. We performed new experiments to better compare the status of immune cell activation between Tie2-IFN α mice and Tie2-GFP mice. As shown in Appendix Fig S5 of the revised version, the total number of splenic CD4⁺ T cells was very similar between Tie2-IFN α mice and Tie2-GFP mice sacrificed at 30 days post-transplantation (Appendix Fig S5D). We also observed in the former mice a slight increase in the number of splenic CD8⁺ T cells that was accompanied by a commensurate increase in markers of T cell activation and markers of central memory (Appendix Fig S5E). In keeping with what we previously reported (De Palma et al, 2008; Escobar et al, 2014), these results indicate that the status of immune cell activation between the two groups of mice is quite comparable, with a general trend of higher CD8⁺ T cell with an activated phenotype in Tie2-IFN α mice. New text describing these results has been added on page 16 lines 16-23 and page 17 lines 1-2 of the revised version.

Response to Referee #3:

1)As the authors rightfully discuss in the ms, their model of performing cell/gene therapy on

naïve mice well before the injection metastatic cells has very little to do with the treatment of established (refractory) metastases, which is also the most common setting for the testing of novel therapies. Indeed, the mouse model used is somewhat more faithful to the adjuvant setting, which the authors propose is a good option for their treatment strategy. Adjuvant trials are performed after curative resection and extensive chemotherapy, neither of which are tested in the current study. To strengthen the rationale for adjuvant trials, the authors should do one or both of the following experiments:

- a) inject CRC cell lines in the colon/caecum before cell/gene therapy and score metastases with IFN γ or control TEMs;*
- b) same injection as in (a) but treating with chemo and/or resecting primary tumours before starting the cell/gene therapy.*

We thank the Referee for his/her comments. Indeed, it would be great to verify the benefit of our Tie2-IFN α approach in immunocompetent mouse models that recapitulate additional aspects of CRC progression in humans; for instance, mice in which we can monitor the relative growth of liver metastases that arise after chemotherapy or surgical resection of primary tumors in the colon or, alternatively, liver metastases that form after direct injection of CRC cells in the large intestine. Unfortunately, mouse models of spontaneous CRC capable of producing liver metastases do not exist and the injection of CRC cells in the cecal wall followed by BM transplantation turned out to be not really practical. In fact, we injected the two CRC cell lines into the cecal wall of ten immunocompetent CB6 mice, as previously reported (Bresalier et al, 1987), and we succeeded at promoting the development of orthotopic CRC in about 60% of the mice. In agreement with previous publications (Bresalier et al, 1987; Zhang et al, 2013), however, by the time (i.e. ~45 days after injection) that we were forced to euthanize the animals for ethical reasons (note that the intestinal tumors became large and the general health of the animals rapidly deteriorated), none of the tumor-bearing mice showed liver metastatic foci.

As also pointed out in our responses to referees 1 and 2, we decided to examine the effect of our Tie2-IFN α approach on liver metastases that are already established in the following way: fifteen mice were injected with a small number of CRC cells (5×10^3 CT26-RFP) directly beneath the Glisson's capsule in a localized area of the left hepatic lobe (Fig 5A and C of the revised version). Eight days after injection, the mice were divided in two groups and subjected to bone marrow (BM) transplantation with HSPCs previously transduced to produce either Tie2-GFP or Tie2-IFN α cells. Four days later (a time point in which Tie2-GFP or Tie2-IFN α cells are not yet emerged from the BM (Lechman et al, 2012; Zonari et al, 2013), the animals were subjected to the first MRI analysis, which - as expected - detected no significant differences in the total tumor volume from the liver of the two groups of mice. Note that at this time point post transplant mice could be safely transferred to the MRI facility in compliance with our biosafety protocols. Follow-up MRI analyses at days 21 and 28 post-transplant revealed that Tie2-IFN α mice display reduced volumes of CRC liver metastases when compared to those detected in Tie2-GFP mice (Fig 5B and C of the revised version). This difference felt short of being statistically significant, probably because of the partial BM reconstitution at these time points (note that at day 28 after BM transplantation, peripheral WBCs were still about 30% below normal counts, Appendix Fig S5A). Remarkably, Tie2-IFN α mice showed a statistically significant increase - over Tie2-GFP mice - of *Irf7*, a prototypical IFN-stimulated gene of the liver (Fig 5F) and a statistically significant improvement in overall survival (Fig 5H of the revised version). These results were accompanied by a statistically significant reduction in the % of Tie2-IFN α mice that develop peritoneal lesions (a complication of Glisson's capsule infiltration with CRC cells spreading into the peritoneal cavity). Indeed, less than 30% of Tie2-IFN α mice showed MRI-based evidence of peritoneal carcinomatosis at day 28, while 100% of Tie2-GFP mice developed such complication at this same time point (Fig 5D and Movie EV8 of the revised version). Text regarding these results has been added in the revised manuscript at pages 15-17.

2) The mechanism of action is not well developed. In Figure 4, the authors quantify cell types in a single time point where the tumour burden differs ~50-fold (Figure 3C, % RFP; one of the conditions has hardly any tumour cells), making the comparison very skewed.

Thus, it is unclear what the analysis contributes to a better understanding. In Figure 5, it is necessary to show that the WT reconstituted BM in irradiated IFN α / β R $^{-/-}$ animals is successful and fully functional, in order to conclude that no BM-derived cell is responsible for the IFN γ -mediated tumour rejection.

We agree with the comment that quantifying cell types in a single time point provides limited information and, therefore, we integrated the original results with additional quantitative data obtained at day 3 after CRC cell injection. As shown in the new Fig EV4 of the revised version, immunohistochemical analyses revealed a comparable increase - over NaCl-injected Tie2-GFP mice - in the number of F4/80 $^{+}$ macrophages as well as CD3 $^{+}$ T cells and B220 $^{+}$ B cells in the liver of both Tie2-GFP mice and Tie2-IFN α mice at day 7 but not at day 3 post-CT26-RFP cell injection (Fig EV4C, D and E). This suggests that CRC cell expansion (more abundant in Tie2-GFP mice) and IFN α release (more abundant in Tie2-IFN α mice) both promoted the intrahepatic recruitment and/or expansion of immune cells, reaching at day 7 a similar overall effect. Note that at this time point, the ratio of recruited immune cells per tumor cell was much higher in Tie2-IFN α mice. Text regarding these results has been added in the revised manuscript at pages 11 lines 19-23.

Regarding the request to show that "WT reconstituted BM in irradiated IFN α / β R $^{-/-}$ animals is successful and fully functional", we would like to point out that all of the chimeric animals from the three groups depicted in the Appendix Fig S4C of the revised version show normal CBC counts, which are indicative of a normal hematopoiesis. This is consistent with previous work from our laboratory where we subjected similar BM chimeras to experimental infections with noncytopathic or cytopathic viruses (i.e. LCMV or VSV) could mount a virus specific adaptive immune response similar to controls and displayed normal capacity to clear the respective virus (Iannacone et al, 2010; Iannacone et al, 2008).

3) As the data may indicate that the effect of IFN γ is not expected to require Kupffer cells, BM-derived cells or to be occurring within the cancer cells themselves, it remains unclear where it is signaling and to what end. The authors should do IHC/immune fluorescence stainings for downstream signaling events (e.g. pSTAT1) to help identify the cell types that may be responsible for the observed effects.

We agree with the referee and followed his/her suggestion. We performed immunohistochemical analyses on liver sections from the same mice described in Fig 3 (of the original and revised version) looking at a staining specific for the phosphorylated form of STAT1. The analyses revealed that at 5 min post saline injection pSTAT1 is already expressed uniformly and at rather low levels in the nucleus of parenchymal and nonparenchymal cells of the liver of Tie2-GFP mice or Tie2-IFN α mice (Figure below, top panels). By day 7 post CRC cell injection, numerous cells became strongly positive for pSTAT1 in both groups of mice, most of which were CRC cells, non-parenchymal cells and hepatocytes (Figure below, middle panels).

As pSTAT1 in our setting appears to be highly expressed by many different cell types independently of Tie2-IFN α , and since pSTAT1 and also other IFN α / β receptor downstream signaling molecules (e.g. pStat2) are not exquisitely specific for IFN α / β receptor signaling, these results don't help in the definitive identification of the cell/cells responsible for the IFN α mediated antitumor responses. Future work attempting to selectively eliminate the IFN α / β receptor on radio-resistant liver cells other than KCs (e.g. endothelial cells, fibroblasts, stellate cells and hepatocytes) *in vivo*, will eventually identify which radio-resistant liver cells are ultimately targeted by this pleiotropic cytokine.

Rebuttal letter References

Bresalier RS, Hujanen ES, Raper SE, Roll FJ, Izkowitz SH, Martin GR, Kim YS (1987) An animal model for colon cancer metastasis: establishment and characterization of murine cell lines with enhanced liver-metastasizing ability. *Cancer Res* 47: 1398-1406

De Palma M, Mazzieri R, Politi LS, Pucci F, Zonari E, Sitia G, Mazzoleni S, Moi D, Venneri MA, Indraccolo S et al (2008) Tumor-targeted interferon-alpha delivery by Tie2-expressing monocytes inhibits tumor growth and metastasis. *Cancer Cell* 14: 299-311

De Palma M, Venneri MA, Galli R, Sergi Sergi L, Politi LS, Sampaolesi M, Naldini L (2005) Tie2 identifies a hematopoietic lineage of proangiogenic monocytes required for tumor vessel formation and a mesenchymal population of pericyte progenitors. *Cancer Cell* 8: 211-226

Escobar G, Moi D, Ranghetti A, Ozkal-Baydin P, Squadrito ML, Kajaste-Rudnitski A, Bondanza A, Gentner B, De Palma M, Mazzieri R et al (2014) Genetic engineering of hematopoiesis for targeted IFN-alpha delivery inhibits breast cancer progression. *Sci Transl Med* 6: 217ra213

Iannacone M, Moseman EA, Tonti E, Bosurgi L, Junt T, Henrickson SE, Whelan SP, Guidotti LG, von Andrian UH (2010) Subcapsular sinus macrophages prevent CNS invasion on peripheral infection with a neurotropic virus. *Nature* 465: 1079-1083

Iannacone M, Sitia G, Isogawa M, Whitmire JK, Marchese P, Chisari FV, Ruggeri ZM, Guidotti LG (2008) Platelets prevent IFN-alpha/beta-induced lethal hemorrhage promoting CTL-dependent clearance of lymphocytic choriomeningitis virus. *Proc Natl Acad Sci U S A* 105: 629-634

Kobaek-Larsen M, Thorup I, Diederichsen A, Fenger C, Hoitinga MR (2000) Review of colorectal cancer and its metastases in rodent models: comparative aspects with those in humans. *Comp Med* 50: 16-26

Lechman ER, Gentner B, van Galen P, Giustacchini A, Saini M, Boccalatte FE, Hiramatsu H, Restuccia U, Bachi A, Voisin V et al (2012) Attenuation of miR-126 activity expands HSC in vivo without exhaustion. *Cell Stem Cell* 11: 799-811

Taketo MM, Edelman W (2009) Mouse models of colon cancer. *Gastroenterology* 136: 780-798

Venneri MA, De Palma M, Ponzoni M, Pucci F, Scielzo C, Zonari E, Mazzieri R, Doglioni C, Naldini L (2007) Identification of proangiogenic TIE2-expressing monocytes (TEMs) in human peripheral blood and cancer. *Blood* 109: 5276-5285

Zhang Y, Davis C, Ryan J, Janney C, Pena MM (2013) Development and characterization of a reliable mouse model of colorectal cancer metastasis to the liver. *Clin Exp Metastasis* 30: 903-918

Zonari E, Pucci F, Saini M, Mazzieri R, Politi LS, Gentner B, Naldini L (2013) A role for miR-155 in enabling tumor-infiltrating innate immune cells to mount effective antitumor responses in mice. *Blood* 122: 243-252

We wish to thank you and the referees for helping us to improve this paper and we hope you agree that the revised manuscript is now acceptable for publication in *EMBO Molecular Medicine*.

2nd Editorial Decision

18 November 2015

Thank you for the submission of your revised manuscript to EMBO Molecular Medicine. We have now received the enclosed reports from the referees that were asked to re-assess it. Unfortunately, Reviewer #2 was not available and therefore I have decided to proceed based on the other two evaluations that in my opinion provide sufficient feedback to fully evaluate your revised manuscript

As you will see, although the reviewers are now globally supportive, Reviewer 3 has a few remaining concerns that I would ask you to deal with. Firstly, s/he would like substantial clarification regarding the apparent spread of secondary lesions in the liver and if this is the case, the effect of TEM-expressed IFN α on them. The reviewer also lists a few other important points that require discussion and/or clarification.

Depending on the completeness of your replies, I might consider making an editorial decision on your next, final version.

In the likely event of acceptance, you will be asked to fulfill a number of editorial requirements as

listed below. I suggest that you provide the following information and amendments requested directly with the next, final version of your manuscript to shorten processing times:

1) As per our Author Guidelines, the description of all reported data that includes statistical testing must state the name of the statistical test used to generate error bars and P values, the number (n) of independent experiments underlying each data point (not replicate measures of one sample), and the actual P value for each test (not merely 'significant' or 'P < 0.05'). If you prefer so, you may collect all the P values in a separate table, which however should be duly referenced to in the manuscript.

2) We are now encouraging the publication of source data, particularly for electrophoretic gels and blots, with the aim of making primary data more accessible and transparent to the reader. Would you be willing to provide a PDF file per figure that contains the original, uncropped and unprocessed scans of all or at least the key gels used in the manuscript? The PDF files should be labeled with the appropriate figure/panel number, and should have molecular weight markers; further annotation may be useful but is not essential. The PDF files will be published online with the article as supplementary "Source Data" files. If you have any questions regarding this just contact me.

3) Every published paper now includes a 'Synopsis' to further enhance discoverability. Synopses are displayed on the journal webpage and are freely accessible to all readers. They include a short standfirst as well as 2-5 one sentence bullet points that summarise the paper. Please provide the synopsis including the short list of bullet points that summarise the key NEW findings. The bullet points should be designed to be complementary to the abstract - i.e. not repeat the same text. We encourage inclusion of key acronyms and quantitative information. Please use the passive voice. Please attach this information in a separate file or send them by email, we will incorporate it accordingly. You are also welcome to suggest a striking image or visual abstract to illustrate your article. If you do please provide a jpeg file 550 px-wide x 400-px high.

4) Please take out the line numbering and the underlining from the manuscript file as they are no longer needed.

5) I would like to suggest a slightly modified title to increase immediate impact. Would the following be to your liking: "IFN gene/cell therapy curbs colorectal cancer colonization of the liver by acting on the hepatic microenvironment"?

Please submit your revised manuscript within two weeks or sooner.

I look forward to reading your revised manuscript.

***** Reviewer's comments *****

Referee #1 (Comments on Novelty/Model System):

The authors did a good job in addressing all the comments and the article is now good for publication.

Referee #3 (Remarks):

Key issues with the initial submission were 1) non-ideal model system to address the clinical setting; and 2) lack of mechanism. To address the first issue, the authors have added experiments with established liver lesions and as to the second, they performed stainings of pStat1 (as suggested) that display a broad tumoral presence of activated Stat1 molecules, independent on IFN α delivery.

Issue 1) The new experiments with established liver lesions, together with the rest of the data,

support the claim that TEM-delivered IFN α has a potential therapeutic benefit for patients with metastatic CRC. In my opinion, they are good enough for publication, with a few points left to be addressed:

1-1) It seems reasonable to assume that (part of the reason why) the authors decided not to use an intrasplenic injection to establish liver mets to then treat with BM-Tie2-IFN α therapy due to the relatively short survival period. I would ask that the authors explain their reason for changing the setting in the ms. Within my interpretation, intrahepatic injection might prolong the therapeutic window because it creates only one tumour at the site of injection. It is therefore confusing to see multiple lesions in Figure 5C: this needs to be addressed/explained. Moreover, if the tumour cells indeed do (surprisingly) spread from the site of injection to form secondary lesions, authors should report whether this spread, i.e. number of lesions, is affected by TEM-expressed IFN α .

1-2) The supplementary methods (Appendix, page 13) mention MC38 cells for the intrahepatic injection, whereas all the other mentions to this experiment (merged document pages 15, 24, 45) state CT26 cells. Perhaps the appendix/Supl Methods is mistaken?

1-3) Also for Figure 5, it is not obvious that the stainings of GFP and Cd11b overlap. The authors should consider showing the GFP/Cd11b merge, next to the 3-colour merge. (They can leave out Hoechst alone.) Alternatively, they can use stainings for MMR or F4/80 as in figure 1 for increased clarity and consistency.

Issue 2) The relatively moderate level of mechanistic insight that this study provides, can be considered outweighed by the potential in therapeutic benefit. Especially when recognizing the difficulties to improve this situation with reasonable experiments and without embarking on a tangential study. Nevertheless, considering the current data, there is one issue concerning (now) figure 4: the conclusion drawn by the authors is not fully supported by the data.

2-1) Fig 4B shows that in the case of IFN α /bR $^{-/-}$ donated TEMs, the metastases are only *partially* inhibited from growing (from ~1100 to 400 mm 3 , cf the ratio of 400/35 in the WT BM donation), indicating that at least some of the IFN α functionality resides in BM-derived cells, while the remaining part may be attributed to radio-resistant resident cells. Therefore, rather than concluding that "Liver radio-resistant cells are primary targets of the antitumor activity of IFN α " (merged document page 12, line 14), the ms should acknowledge the possibility of BM-derived cells and even TEMs themselves being the functional mediators in addition to unknown resident microenvironmental agents.

2-2) Although the pStat1 antibody was used in the rebuttal, it does not appear in the revision (merged manuscript) and needs not be added in its methods section, page 25 (or appendix, page 18).

2nd Revision - authors' response

25 November 2015

Thank you for your letter regarding our revised manuscript we submitted to EMBO Molecular Medicine. As suggested we modified the title into "**IFN α gene/cell therapy curbs colorectal cancer colonization of the liver acting on the hepatic microenvironment**". Further, we added actual p-value for each test in the corresponding figure and prepared the synopsis and a striking image that we will send you by email. We also thank referee #2 for his/her constructive comments. We have thus revised our manuscript accordingly and believe that our manuscript is now good for publication.

Response to Referee #1:

The authors did a good job in addressing all the comments and the article is now good for publication.

We thank the referee for his/her positive evaluation of our work.

Response to Referee #2:

1-1) It seems reasonable to assume that (part of the reason why) the authors decided not to use an intrasplenic injection to establish liver mets to then treat with BM-Tie2-IFN α therapy due to the relatively short survival period. I would ask that the authors explain their

reason for changing the setting in the ms.

Within my interpretation, intrahepatic injection might prolong the therapeutic window because it creates only one tumour at the site of injection. It is therefore confusing to see multiple lesions in Figure 5C: this needs to be addressed/explained.

Moreover, if the tumour cells indeed do (surprisingly) spread from the site of injection to form secondary lesions, authors should report whether this spread, i.e. number of lesions, is affected by TEM-expressed IFN α .

We thank the referee for his/her comment, indeed we chose the intrahepatic injection route to minimize CRC cell spreading within the liver, and thus allowing mice to reconstitute their BM in the presence of few established and fast growing hepatic lesions. We also chose this approach because mice do not necessitate splenectomy thus reducing morbidity and possible confounding factors. Regarding the issue of intrahepatic spreading, we would like to point out that this route of injection produced a limited number of hepatic lesions that were confined to the point of injection and to the path of the injecting needle. As suggested we added on page 15 the number of MRI detectable lesions 4 days after BM transplantation, that were indeed very similar between the two groups (mean number of lesions in Tie2-GFP=1.5 \pm 0.4; Tie2-IFN α =2 \pm 0.28). Furthermore MRI analyses at days 21-28 revealed again similar numbers of intrahepatic lesions (mean number of lesions at day 21-28: Tie2-GFP=3 \pm 0.53; Tie2-IFN α =3.12 \pm 0.63), however Tie2-IFN α mice display reduced volumes of CRC liver metastases when compared to those detected in Tie2-GFP mice. For the seek of clarity we drew red dashed lines to indicate hepatic lesions and green dashed lines to indicate extrahepatic lesions consistently with Movie EV8. These data indicate that intrahepatic spreading is limited to the path of the injecting needle and that Tie2-IFN α therapy is able to reduce tumor growth and secondary intraperitoneal spreading as indicated by data displayed on Fig 5D. New text was added on page 15 and 24 of the revised version.

1-2) The supplementary methods (Appendix, page 13) mention MC38 cells for the intrahepatic injection, whereas all the other mentions to this experiment (merged document pages 15, 24, 45) state CT26 cells. Perhaps the appendix/Supl Methods is mistaken?

We thank the referee and corrected the mistake in the Appendix, page 14.

1-3) Also for Figure 5, it is not obvious that the stainings of GFP and Cd11b overlap. The authors should consider showing the GFP/Cd11b merge, next to the 3-colour merge. (They can leave out Hoechst alone.) Alternatively, they can use stainings for MMR or F4/80 as in figure 1 for increased clarity and consistency.

We thank the referee for his/her comment, and we included higher magnification insets in the merge panels of Fig 5E to highlight GFP and Cd11b signal overlap.

*2-1) Fig 4B shows that in the case of IFN α /bR $^{-/-}$ donated TEMs, the metastases are only *partially* inhibited from growing (from ~1100 to 400 mm 3 , cf the ratio of 400/35 in the WT BM donation), indicating that at least some of the IFN α functionality resides in BM-derived cells, while the remaining part may be attributed to radio-resistant resident cells. Therefore, rather than concluding that "Liver radio-resistant cells are primary targets of the antitumor activity of IFN α ;" (merged document page 12, line 14), the ms should acknowledge the possibility of BM-derived cells and even TEMs themselves being the functional mediators in addition to unknown resident microenvironmental agents.*

We agree with the referee comment and further discussed the possibility that also BM derived cells, including TEMs, indeed may participate in the anti-tumoral process induced by Tie2-IFN α . We would also like to point out that results displayed in Fig 4B - where we transplanted C57BL/6 HSPCs into IFN α / β R $^{-/-}$ mice - showed no effect of Tie2-IFN α therapy when radio-resistant hepatic cells do not have the genetic capacity to sense IFN α , thus the radio-resistant hepatic microenvironment plays a primary role in this process. New text on this issue was added on page 14.

2-2) Although the pStat1 antibody was used in the rebuttal, it does not appear in the

revision (merged manuscript) and needs not be added in its methods section, page 25 (or appendix, page 18).

We thank the referee and corrected the mistake in the manuscript and Appendix.

Finally, we wish to thank you and the referees for helping us to improve this paper and we hope you agree that the revised manuscript is now acceptable for publication in EMBO Molecular Medicine.